# Structural dynamics of RAF1-HSP90-CDC37 and HSP90 complexes reveal asymmetric client interactions and key structural elements

Lorenzo I. Finci[1], Mayukh Chakrabarti [1,5], Gulcin Gulten[1,5], Joseph Finney[2], Carissa Grose[1], Tara Fox[2], Renbin Yang[3], Dwight V. Nissley [1], Frank McCormick [1,4], Dominic Esposito [1], Trent E. Balius [1] & Dhirendra K. Simanshu [1] ✉

RAF kinases are integral to the RAS-MAPK signaling pathway, and proper RAF1 folding relies on its interaction with the chaperone HSP90 and the cochaperone CDC37. Understanding the intricate molecular interactions governing RAF1 folding is crucial for comprehending this process. Here, we present a cryo-EM structure of the closed-state RAF1-HSP90-CDC37 complex, where the C-lobe of the RAF1 kinase domain binds to one side of the HSP90 dimer, and an unfolded N-lobe segment of the RAF1 kinase domain threads through the center of the HSP90 dimer. CDC37 binds to the kinase C-lobe, mimicking the N-lobe with its HxNI motif. We also describe structures of HSP90 dimers without RAF1 and CDC37, displaying only N-terminal and middle domains, which we term the semi-open state. Employing 1 μs atomistic simulations, energetic decomposition, and comparative structural analysis, we elucidate the dynamics and interactions within these complexes. Our quantitative analysis reveals that CDC37 bridges the HSP90-RAF1 interaction, RAF1 binds HSP90 asymmetrically, and that HSP90 structural elements engage RAF1's unfolded region. Additionally, N- and C-terminal interactions stabilize HSP90 dimers, and molecular interactions in HSP90 dimers rearrange between the closed and semi-open states. Our findings provide valuable insight into the contributions of HSP90 and CDC37 in mediating client folding.

The RAF kinases, ARAF, BRAF, and RAF1, are members of the serine/threonine kinase family[1,2]. Their activation by RAS proteins at the membrane initiates MAPK signaling, regulating cell growth, division, and survival. The N-terminal domain of RAF kinases consists of a RAS-binding domain, a cysteine-rich domain, and a phosphorylation-dependent binding site for 14-3-3 proteins, a family of scaffolding molecules that modulate the function of other proteins through phosphorylation-dependent mechanisms[1,2] (Fig. 1a). The C-terminal domain contains a kinase domain and another 14-3-3 binding site. The RAF1 kinase domain (KD; residues 349-609) is composed of an N-lobe (residues 349-424) and a C-lobe

(residues 429-609), connected by a flexible linker. The N-lobe comprises five antiparallel β-strands and the regulatory αC helix, while the C-lobe consists of α-helices. The kinase active site, which binds nucleotides (ADP/ATP) and magnesium, is located at the interface of these two lobes. The proper folding and maturation of many kinases, including RAF1, requires chaperones and co-chaperones. The stability, subcellular localization, activity, and MAPK signaling of RAF1 depend on its interaction with molecular chaperone HSP90[3]. HSP90 employs ATP hydrolysis to aid the folding and maturation of diverse client proteins within a multi-chaperone complex involving the kinase-specific co-chaperone CDC37.

[1]NCI RAS Initiative, Cancer Research Technology Program, Frederick National Laboratory for Cancer Research, Frederick, MD, USA. [2]National Cryo-EM Facility, Cancer Research Technology Program, Frederick National Laboratory for Cancer Research, Frederick, MD, USA. [3]Center for Molecular Microscopy, Cancer Research Technology Program, Frederick National Laboratory for Cancer Research, Frederick, MD, USA. [4]Helen Diller Family Comprehensive Cancer Center, University of California San Francisco, San Francisco, CA, USA. [5]These authors contributed equally: Mayukh Chakrabarti, Gulcin Gulten. ✉e-mail: dhirendra.simanshu@nih.gov

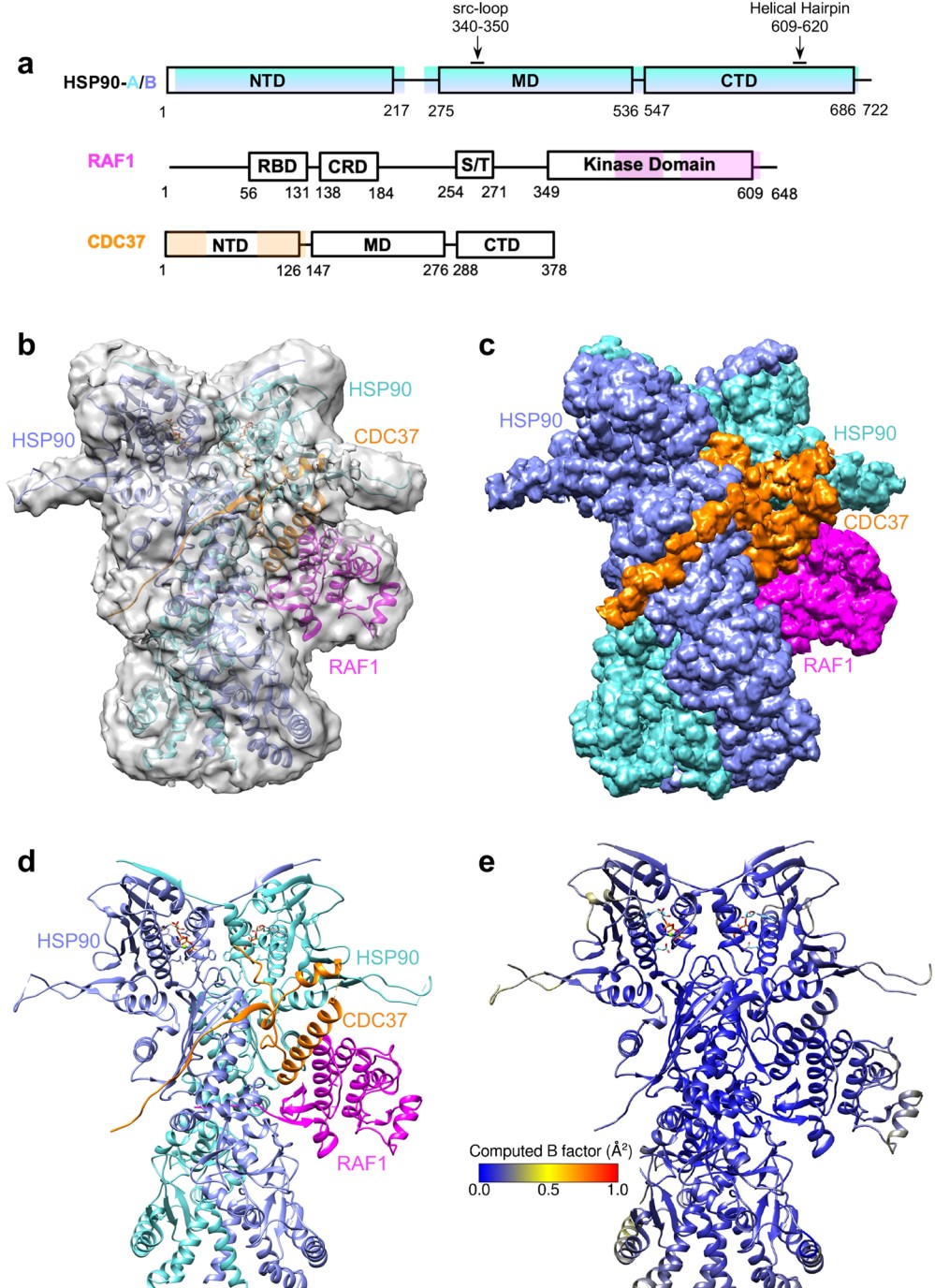

**Fig. 1 | The overall structure of the RAF1-HSP90-CDC37 complex in the closed state. a** The domain architecture of HSP90, RAF1, and CDC37. The regions observed in the map and model corresponding to HSP90, RAF1, and CDC37 are colored blue, magenta, and orange, respectively. Both protomers A and B of HSP90 have the same range of residues visible in the map. **b** The overall structure of the RAF1-HSP90-CDC37 complex showing the map (light gray) associated with the three proteins using the same color scheme as in panel a, except that HSP90 protomers A and B are colored in cyan and blue, respectively. **c, d** The overall structure of the RAF1-HSP90-CDC37 complex shown (**c**) in the map colored by zone, and (**d**) in cartoon representation. **e** The overall structure of the RAF1-HSP90-CDC37 complex in the closed state, colored by RMSF-based B-factor obtained from molecular simulation. The computed B-factor is normalized such that a direct comparison can be made to the semi-open state in Fig. 6f, g.

HSP90 isoforms are present across all eukaryotes and in various cellular compartments. In humans, HSP90α and HSP90β reside in the cytoplasm, while GRP94/GRP96 (glucose-regulated protein) exists in the ER, and HSP75/TRAP1 (tumor necrosis factor receptor-associated protein) is found in mitochondria[4]. Similar family members are present in different species: HSP84/HSP86 in mice, HSP82/HSC82 in yeast, and HSP83 in *Drosophila* and *Trichoplusia ni*[4,5]. HSP90 and its homologs function as

dimers[6], wherein each protomer has three structural domains (Fig. 1a). The N-terminal domain (NTD) has a nucleotide (ATP/ADP) binding site. The middle domain (MD) is implicated in client binding and contains an amphipathic src-loop[7]. The C-terminal domain (CTD) contains the main dimerization interface for forming the HSP90 dimer and includes an amphipathic helical hairpin (HH). In the absence of ATP and client, HSP90 mostly exists in an open "V-shaped" conformation anchored by the

C-terminal dimerization interface. ATP binding to the NTD promotes the formation of a closed state primed for ATP hydrolysis, in which the HSP90 NTDs transiently dimerize and induce structural rearrangement of the MD, while the protomers remain associated through their CTDs[8,9]. Conformational changes during this process involve β-strand swapping and α-helix movement, facilitating stabilizing interactions and enabling transient dimerization[10–13].

CDC37 facilitates client protein recruitment to HSP90 by interacting with the client and HSP90 to aid their association[14,15]. CDC37 comprises three domains: an N-terminal domain binding the client, a middle domain engaging HSP90, and an extended C-terminal domain for CDC37 dimerization (Fig. 1a). While a model for the HSP90-CDC37-mediated client kinase maturation cycle has been proposed, some molecular details remain elusive. According to the model, CDC37 initially binds a partially unfolded client, followed by CDC37-client association with the open-state HSP90 dimer. Subsequent ATP binding induces structural changes, forming a closed-state ATP-bound HSP90-CDC37-client complex. Protein phosphatase 5 dephosphorylates a phosphoserine residue (S13) on CDC37 within this complex, leading to CDC37 dissociation. Upon ATP hydrolysis, the client-bound HSP90 complex releases the properly folded client, while HSP90 returns to its open state[16].

Several earlier studies have improved our understanding of the intricacies of chaperone-client interactions[17,18]. The cryogenic electron microscopy (cryo-EM) structure of the CDK4-HSP90-CDC37 complex provided insight into how HSP90 stabilizes an unfolded client[19]. In this structure, the CDK4 kinase adopts an inactive conformation, wherein the N- and C-lobes of the kinase domain are split, each residing on opposite sides of a closed HSP90 dimer. The unfolded strand from the kinase N-lobe threads through the lumen, an interfacial cavity formed at the dimer interface between two HSP90 protomers.

In this work, we solved the closed state structure of RAF1 bound to HSP90 and CDC37 by cryo-EM. The structure reveals the RAF1 KD C-lobe, with an unfolded segment of the N-lobe threading through the lumen of the HSP90 dimer. The N-terminal domain of CDC37 binds the kinase C-lobe, mimicking the interactions of the kinase N-lobe. These characteristics resemble interactions observed in the structure of CDK4-HSP90-CDC37. During the preparation of this manuscript, structures of HSP90 and CDC37 in complex with RAF1[20], or in complex with BRAF$_{V600E}$[21], with and without protein phosphatase 5[21,22], were reported. Nonetheless, multiple independent structural analyses of RAF1 and BRAF$_{V600E}$ complexed with HSP90 and CDC37 would help in understanding the role of structural elements in HSP90 and CDC37, which may be implicated in conserved modes of recognition of the RAF kinases. Additionally, by characterizing molecular interactions between HSP90, the client, and the cochaperone in the closed state using molecular simulations, we provide further insight into how HSP90 achieves the delicate balance between promiscuity and specificity. Using recently reported cryo-EM structures of HSP90 complexes with various clients, we conducted a comparative structural analysis of the unfolded regions of clients threaded through the luminal cavity created by the association of HSP90 protomers. This analysis permits us to discuss prevailing hypotheses about determinants of client folding by the HSP90-CDC37 complexes. We distinguish that the helical hairpins (CTD) and src-loops (MD) of HSP90 frame the luminal cavity, a preserved structural feature among all analyzed structures.

During 3D particle classification, we observed a substantial population of HSP90 in a more open conformation, lacking RAF1 and CDC37. To characterize this population, we solved structures of HSP90 in which the CTDs are unresolvable, dimerization occurs by the NTDs bound to ATP (ADP + molybdate), and in which most of the MD is resolved and participates in dimerization. We refer to these NTD-MD HSP90 dimers as a "semi-open state". To further support the semi-open state, we also solved a semi-open structure using a cross-linked sample. Additionally, we performed 1 μs of atomistic simulation for the three complexes to obtain insight into molecular interactions characterizing these structures, and analyzed

them using pairwise-residue energetic decomposition. We identify a redistribution of contacts in the NTD-MD HSP90 dimer relative to the closed state complex, quantify asymmetric interactions of HSP90 with RAF1, and note HSP90 NTD interactions that stabilize both the closed and semi-open states. Furthermore, we posit a role for CDC37 in bridging interactions between HSP90 and RAF1.

## Results

### RAF1-HSP90-CDC37 complex formation and cryo-EM analysis

We co-expressed human RAF1 and CDC37 in insect cells and purified the RAF1-HSP90-CDC37 complex, ensuring the presence of only homogeneous insect HSP90 proteins and preventing a mixture of human and insect HSP90s (Supplementary Fig. 1). The insect HSP90, hereafter termed HSP90, closely resembles human HSP90 in length and sequence identity (Supplementary Fig. 2). Characterization of the purified complex suggested a molecular weight of 286 kDa indicating combined molecular weight of two HSP90 protomers and one each from RAF1 and CDC37 (Supplementary Fig. 3).

To overcome issues with preferred orientation and conformational heterogeneity inherent in the initial dataset, we collected additional datasets totaling 14,681 micrographs (1,877,778 particles) (Supplementary Figs. 4, 5). Efforts in processing with a larger particle count enabled the identification of a closed state in 32% and a semi-open state in 35% of the data (Supplementary Fig. 5). Utilizing signal subtraction, focused classification, Bayesian polishing, and CTF refinement, we obtained a 3.7 Å closed state map (Supplementary Fig. 6; see "Methods" for details). A semi-open state map was reconstructed at 3.5 Å (Supplementary Fig. 7). To improve stability and conformational homogeneity, we conducted crosslinking experiments, obtaining a cryo-EM map of the crosslinked semi-open state at 3.9 Å (Supplementary Figs. 8, 9). Cryo-EM data collection, refinement and validation statistics are shown in Table 1.

### Structure of RAF1-HSP90-CDC37 in the closed state

Our final model of the RAF1-HSP90-CDC37 complex consists of an HSP90 dimer, the C-lobe of the RAF1 KD, and portions from the N-terminal and middle domains of CDC37 (Fig. 1b–d). Although the RAS-binding domain, cysteine-rich domain, and kinase N-lobe of RAF1, along with the middle and C-terminal domains of CDC37, were present in the purified complex containing full-length proteins, they were not visible in the density maps, suggesting their dynamic behavior in the closed state. The C-lobe of the RAF1 KD is located on one side of the HSP90 dimer, forming both luminal and peripheral interactions with HSP90.

In the RAF1-HSP90-CDC37 complex, HSP90 adopts a nearly 2-fold symmetric structure with a Cα RMSD of 0.7 Å between two protomers, HSP90-A and HSP90-B. The closed state HSP90 dimeric conformation is similar (Cα RMSD of 1.4 Å) to its HSP90 homolog in human CDK4-HSP90-CDC37 complex (PDB: 5FWK). The RAF1 and CDC37 interactions with HSP90 are asymmetric, interacting differently with each HSP90 protomer. The RAF1 C-lobe forms peripheral interactions with both HSP90 protomers and the CDC37 N-terminal domain. In the lumen, the unfolded region of RAF1 N-lobe interacts with two structural elements of both HSP90 protomers: the src-loop (MD) and the helical hairpin (CTD). The CDC37 N-terminal domain interacts with the MD of both HSP90 protomers, and most of these interactions occur with HSP90-B. Based on 1 μs of atomistic simulation of the closed state, the complex remains relatively stable, with larger fluctuations observed only at the periphery of the HSP90 protomers (Fig. 1e).

To assess the folding status of the RAS-binding domain and cysteine-rich domain within the RAF1-HSP90-CDC37 complex, we investigated the KRAS binding affinity of RAF1 in the purified complex. We performed a pull-down assay and observed that avi-tagged KRAS interacted with the closed state complex only when bound to GMPPNP, and not GDP (Supplementary Fig. 10). This observation aligns with a previous study suggesting that RAS can interact with the RAF1-HSP90-CDC37 complex at the membrane[23].

**Table 1 | Cryo-EM data collection, refinement and validation statistics**

| | Closed State (EMD-41816) (PDB 8U1L) | Semi-Open State (EMD-41817) (PDB 8U1M) | Crosslinked Semi-Open State (EMD-41818) (PDB 8U1N) |
|---|---|---|---|
| Data collection and processing | | | |
| Electron Microscope | Titan Krios | Titan Krios | Titan Krios |
| Detector | Gatan K2 | Gatan K2 | Gatan K2 |
| Magnification | 29,000 | 29,000 | 29,000 |
| Voltage (kV) | 300 | 300 | 300 |
| Electron exposure (e–/Å$^2$) | 50 | 50 | 50 |
| Defocus range (μm) | −1.5 to −3 | −1.5 to −3 | −1.5 to −3 |
| Pixel size (Å) | 0.858 | 0.858 | 0.858 |
| Symmetry imposed | C1 | C1 | C1 |
| Initial particle images (no.) | 1,877,778 | 1,877,778 | 579,790 |
| Final particle images (no.) | 97,569 | 325,732 | 139,556 |
| Map resolution (Å) | 3.7 | 3.5 | 3.9 |
| FSC threshold | 0.143 | 0.143 | 0.143 |
| Map resolution range (Å) | 3.34–6.81 | 3.11–6.78 | 3.50–5.21 |
| Refinement | | | |
| Initial model used (PDB code) | Phyre2 model, 5FWK, 3OMV | Phyre2 model | Native Semi-Open state |
| Model resolution (Å) | 3.8 | 3.9 | 3.7 |
| FSC threshold | 0.143 | 0.143 | 0.143 |
| Map sharpening method | Phenix LocalAniso, LocScale | DeepEMhancer, LocScale | DeepEMhancer, LocScale |
| Correlation Coefficient (CCmask) | 0.81 | 0.68 | 0.69 |
| Model composition | | | |
| Non-hydrogen atoms | 12,595 | 7882 | 8164 |
| Protein residues | 1543 | 957 | 994 |
| Ligands | 2:ATP, 2:Mg | 2:ATP, 2:Mg | 2:ATP, 2:Mg |
| B factors (Å$^2$) (min/max/mean) | | | |
| Protein | 30.0/547.2/233.4 | 30.0/465.5/224.4 | 74.1/321.6/162.0 |
| Ligand | 169.4/227.6/191.9 | 175.0/229.9/202.8 | 87.5/101.5/98.8 |
| R.m.s. deviations | | | |
| Bond lengths (Å) | 0.003 | 0.003 | 0.003 |
| Bond angles (°) | 0.717 | 0.613 | 0.658 |
| Validation | | | |
| MolProbity score | 1.70 | 2.21 | 2.06 |
| Clashscore | 14.42 | 23.45 | 17.84 |
| Rotamer outliers (%) | 0.00 | 0.35 | 0.00 |
| CaBLAM outliers (%) | 1.13 | 1.81 | 1.84 |
| Ramachandran plot | | | |
| Favored (%) | 97.83 | 95.04 | 95.54 |
| Allowed (%) | 2.17 | 4.96 | 4.46 |
| Outliers (%) | 0 | 0 | 0 |

**The unfolded RAF1 N-lobe β5-strand threads through the luminal cavity of HSP90**

From the analysis of our closed state map, it is apparent that there is tubular density threading through the HSP90 lumen (Fig. 2a). This luminal cavity is mainly formed from residues comprised of flexible src-loops (MD) and helical hairpins (HH) (CTD) from each HSP90 protomer, as well as residues from α16 (residues 423-443) and α21 (residues 516-524), both from the HSP90 MD. The surface of this luminal cavity is amphipathic, displaying hydrophobic patches (Fig. 2b). The unfolded RAF1 N-lobe β5-strand (residues 418-426) occupies the luminal cavity. The HxN motif of the KD N-lobe contacts the C-lobe in folded RAF1, an interaction replaced by the CDC37

HxN motif in the closed state (Fig. 2c–e). The unfolded β5-strand (Fig. 2b) interacts with residues present on the src-loops and HH of both protomers of HSP90 (Fig. 3a–d), and exhibits amphipathic characteristics that complement the amphipathic nature of the luminal cavity. Through molecular simulations and energetic decomposition analysis, we quantitatively characterized polar (Fig. 3a) and hydrophobic (Fig. 3b) interactions observed structurally between RAF1 and HSP90 residues in the luminal cavity. We observed that the threaded RAF1 β-strand forms asymmetric van der Waals (vdW) interactions with both the src-loop (Fig. 3c) and HH regions (Fig. 3d) of the HSP90 protomers. A recent study also noted asymmetric RAF1-HSP90 interactions[20]. RAF1 only occupies 15% of the volume (824.6 Å$^3$) of the

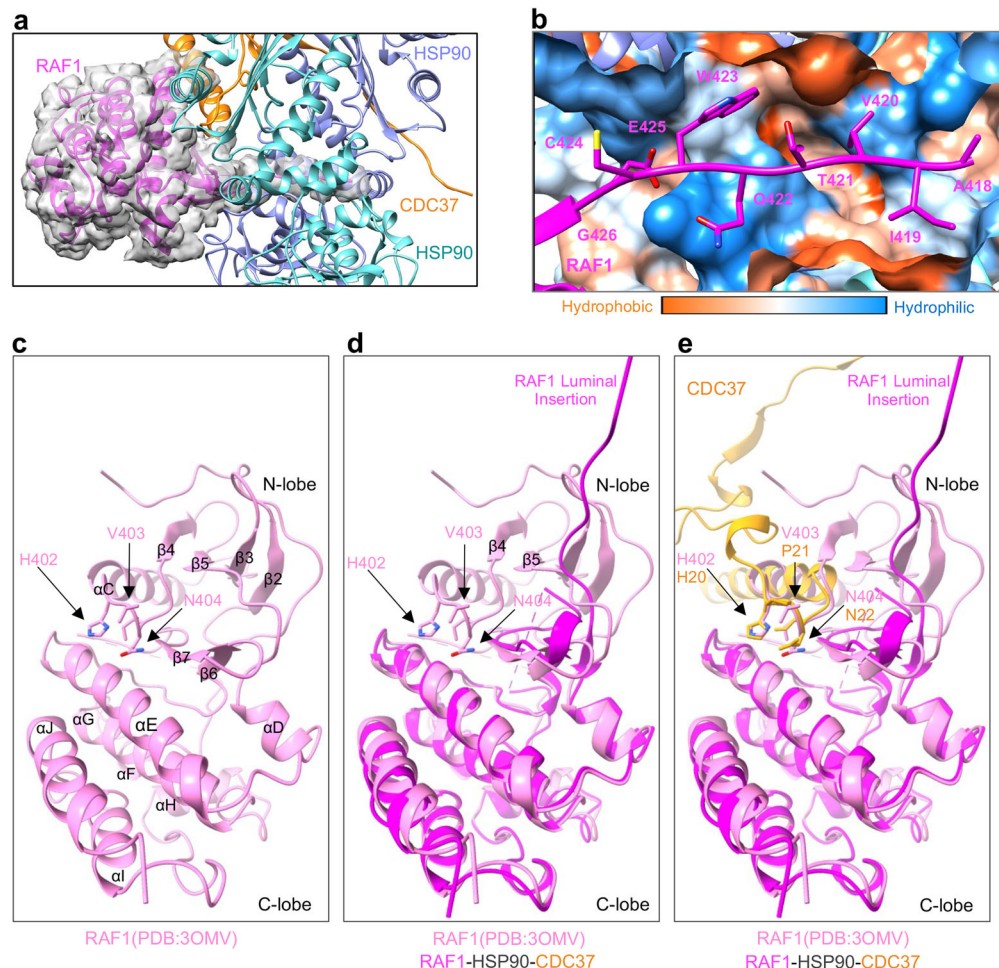

**Fig. 2 | The kinase domain of RAF1 in the RAF1-HSP90-CDC37 complex, and overlay of CDC37 onto the RAF1 kinase domain showing how CDC37 mimics the kinase N-lobe to engage with the kinase C-lobe. a** Enlarged view of the closed-state complex, showing the cryo-EM Coulomb potential map of RAF1 in semi-transparent surface representation. **b** A zoom-in view of the unfolded region of RAF1 present in the luminal cavity of HSP90 in the closed complex. Side chains of RAF1 residues are shown in the stick model, and the two HSP90 protomers and CDC37 are shown in surface representation and colored according to the hydrophobicity and hydrophilicity of the residues. **c** Crystal structure of the folded RAF1 kinase domain (PDB:3OMV, Chain A) showing the interaction between the αC-β4 loop of the N-lobe with the β7 and αE of the C-lobe. The arrows show the location of the HxN residues present in the αC-β4 loop of the N-lobe. **d** The overlay of RAF1 from the crystal structure (PDB:3OMV) and RAF1 in the RAF1-CDC37-HSP90 complex, showing the N-lobe separated from the C-lobe, with the unfolded β5-strand of N-lobe threaded through the luminal cavity of HSP90. **e** The overlay of RAF1 from the crystal structure (PDB:3OMV, Chain A) and RAF1-CDC37 from the RAF1-CDC37-HSP90 complex shows how the HxN motif in CDC37 mimics the HxN motif of the N-lobe while interacting with the β7 and αE of the C-lobe of RAF1.

luminal cavity (5300 Å³) (Fig. 3e and Supplementary Methods). This considerable difference in volumes suggests that other client proteins could interact with this region differently, potentially by engaging in alternative binding interactions with luminal residues, resulting in filling the luminal cavity differently. This hypothesis may explain, in part, how the luminal cavity can accommodate unfolded protein regions with diverse sequences.

**Dynamic interactions in the RAF1-HSP90-CDC37 complex analyzed using molecular simulations**

Considering the variable resolution of the map of the RAF1-HSP90-CDC37 complex, we identified key interfacial interactions. Peripheral interactions between the C-lobe of RAF1 and HSP90 included interactions formed by HSP90-A residues R337, F340, and W311, with the latter two residues previously implicated in client binding[24] (Fig. 4a, b). To gain insight into RAF1-HSP90 interactions, we complemented our structural observations with data obtained from 1 µs of molecular simulation. Molecular snapshots were taken at regular intervals, allowing us to illustrate several intriguing interactions within the complex. These snapshots highlighted side-chain conformational variability, and the

closest and farthest interactions between specific residues (Fig. 4c). In the luminal region, F343 (HSP90-A, src-loop) contacts RAF1 C424 through a backbone-backbone interaction, and RAF1 Q422 through an amino-aromatic interaction. Another notable luminal interaction involves the side chain of E344 (HSP90-B, src-loop), and the backbone atoms of RAF1 A418. Among peripheral interactions, notable salt bridges form between R611 (HH in HSP90-B), and RAF1 residues, E425 and E478. The salt bridge between R611 (HSP90-B) and E425 (RAF1 luminal residue) displayed dynamic fluctuations in strength during the simulation (Fig. 4c). Averaged over the simulations, this electrostatic interaction was the strongest coulombic residue-residue interaction observed between RAF1 and HSP90 (−109.67 kcal/mol). The electrostatic interaction between R611 (HSP90-B) and E478 (RAF1), with an energy of −76.55 kcal/mol, was the second strongest coulombic interaction. Our analysis also suggested favorable vdW contacts between HSP90-A W311 and RAF1 D447 and R450 (Fig. 4c), and a salt bridge interaction between R337 (HSP90-A) and D447 (RAF1) that exhibited a dynamic nature, intermittently breaking and re-forming during the simulation. Interestingly, this salt bridge interaction was not evident in our cryo-EM

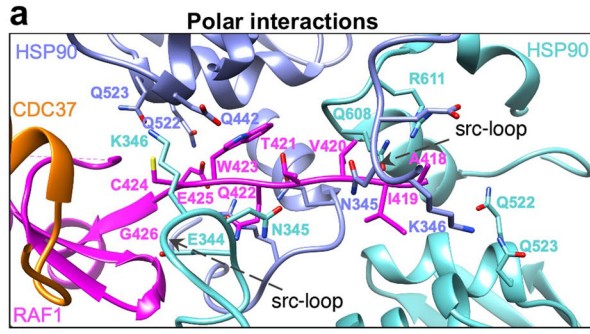

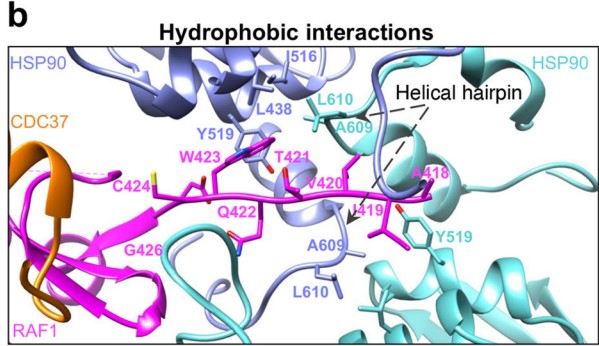

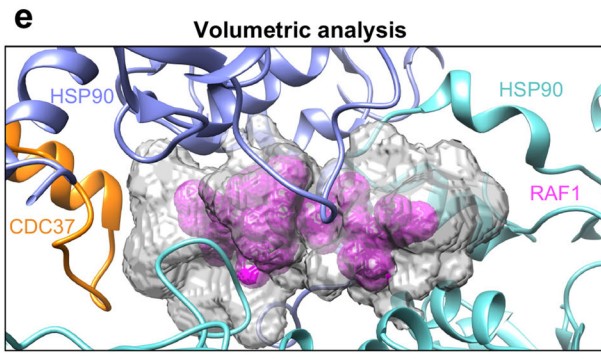

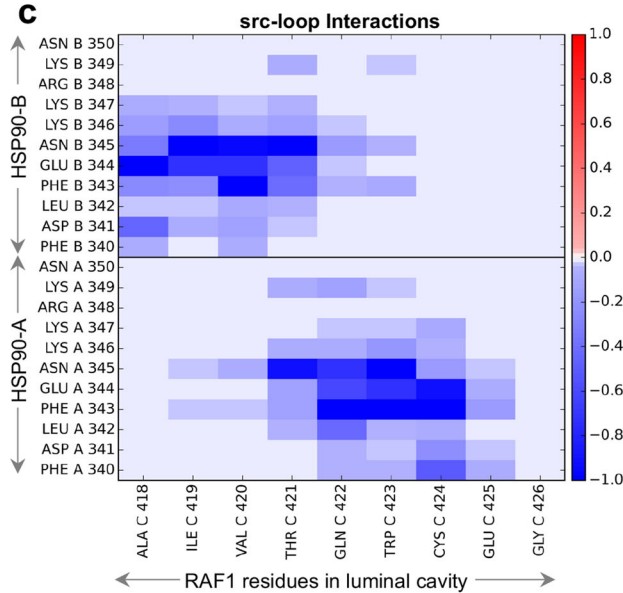

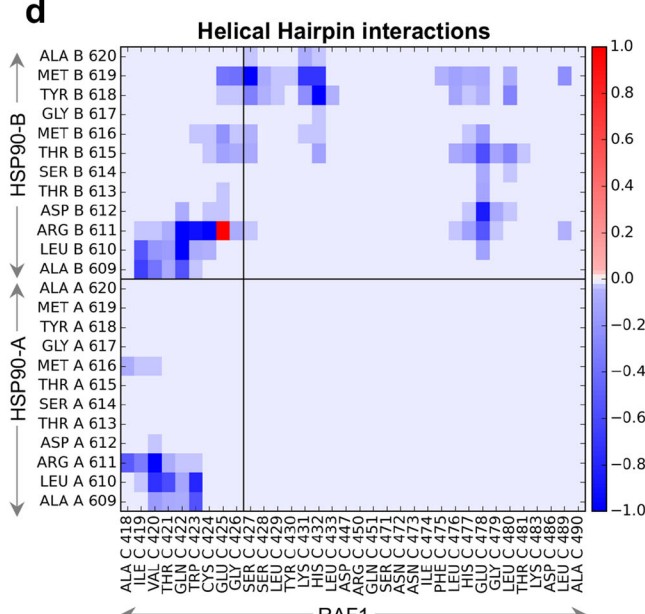

**Fig. 3 | The interactions formed by RAF1 in the luminal cavity formed by HSP90 protomers in the RAF1-HSP90-CDC37 complex. a, b** The same view as Fig. 2b showing interactions formed by (**a**) polar and (**b**) hydrophobic residues of HSP90 with the unfolded region of RAF1 inside the luminal cavity. **c, d** A heat map from molecular simulation energetic decomposition analysis, depicting van der Waals interactions formed by residues of (**c**) src-loop and (**d**) helical-hairpin of HSP90 protomers with RAF1. Horizontal lines delineate interactions formed by two protomers of HSP90, whereas vertical lines delineate interactions formed by RAF1

residues present inside and outside of the luminal cavity of HSP90. Favorable van der Waals contacts between residues are denoted in blue, whereas unfavorable van der Waals contacts are indicated in red (many of these correspond to H-bond and salt bridge interactions). The scale bar is shown for an interaction energy threshold ranging between −1.0 to 1.0 kcal/mol. **e** The volumetric analysis of the luminal cavity formed by HSP90 and the enclosed unfolded region of RAF1. The volume of the luminal cavity is shown in gray, while the volume corresponding to the unfolded region of RAF1 is shown in magenta.

structure, suggesting it may arise due to the dynamic nature of the complex. The electrostatics analysis revealed a value of −60.57 kcal/mol for this salt bridge interaction. Throughout our simulations, we observed strong long-range charge-charge interactions between HSP90 src-loops and RAF1 residues in the luminal cavity. Furthermore, we noticed that short-range interactions were driven by vdW contacts and transient H-bonds, accounting for many weak interactions.

### CDC37 mimics the αC-β4 loop of RAF1 N-lobe and is phosphorylated at Serine13

In the RAF1-HSP90-CDC37 map, we successfully modeled CDC37 residues 2–48 and 102–137 (Fig. 5a). The density associated with the coiled-coil

region (residues 49–101) and beyond residue 137 in CDC37 was too weak to model confidently. Within the RAF1-HSP90-CDC37 complex, CDC37 engages RAF1 in a manner reminiscent of its interaction with CDK4 in the CDK4-HSP90-CDC37 complex, primarily relying on extensive hydrophobic interactions. In the folded RAF1 KD (PDB: 3OMV), the HxN motif present in the N-lobe αC-β4 loop interacts with β7 and αE of the C-lobe. This motif, uniquely conserved in eukaryotic protein kinases[25], is also found in CDC37. The CDC37 HxN motif, a structural element, interacts with the C-lobe β7 and αE by mimicking the RAF1 N-lobe (Figs. 2e and 5b). Notably, the strongest pairwise-residue vdW interaction in the complex occurs between CDC37 residue H20 (HxN motif), and RAF1 Y458 (−3.65 kcal/mol). Our energetic decomposition analysis further reveals vdW

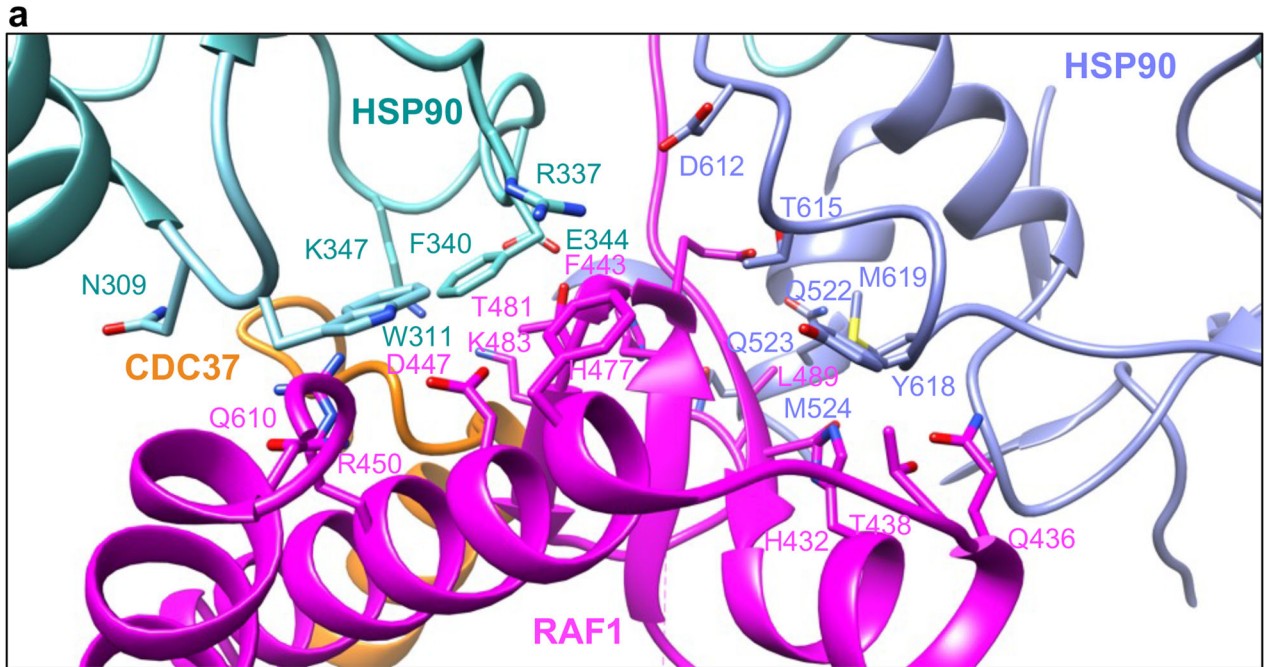

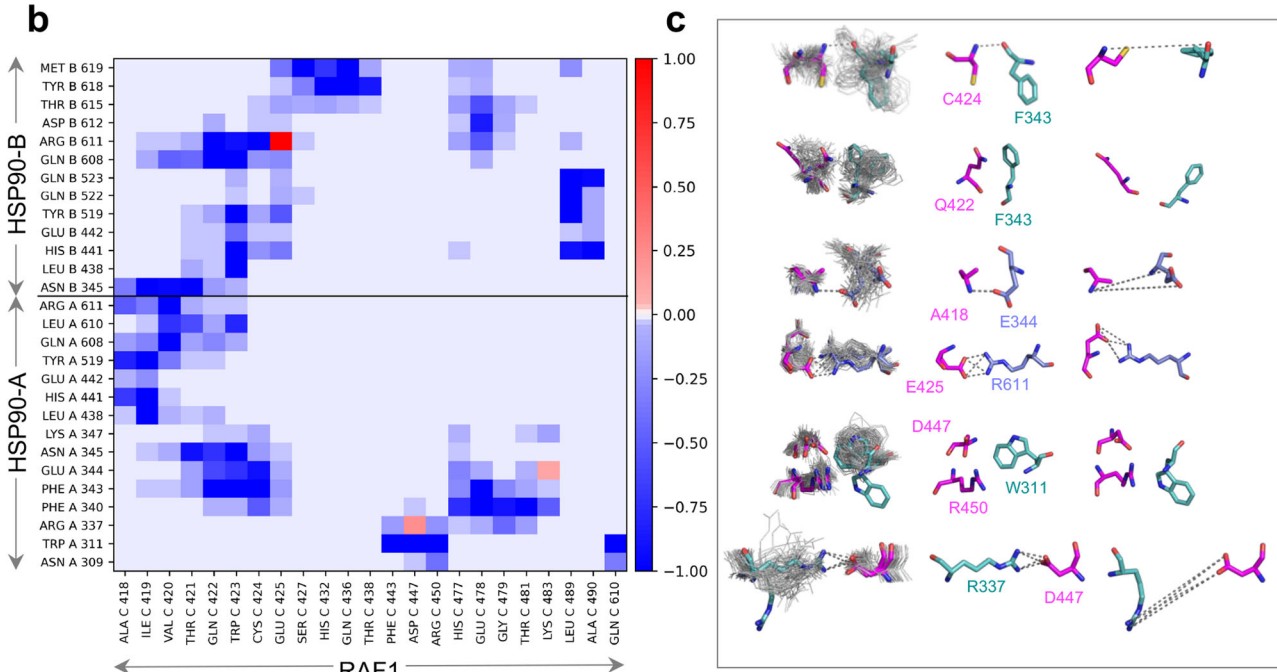

**Fig. 4 | The peripheral interactions formed by RAF1 in the RAF1-HSP90-CDC37 complex. a** Enlarged view showing peripheral interactions formed by RAF1 residues with HSP90 and CDC37 residues in the closed state. **b** Heatmap from molecular simulation energetic decomposition analysis, depicting van der Waals interactions of HSP90 with RAF1. Favorable van der Waals contacts between residues are denoted in blue, whereas unfavorable van der Waals contacts are indicated in red (most of these correspond to salt bridge interactions). The scale bar is shown for an interaction energy threshold ranging between −1.0 to 1.0 kcal/mol. **c** Representative snapshots sampled at equal intervals across the molecular dynamics simulations, illustrating side chain sampling variability of key residues. Gray lines show the side chain conformations sampled; colored sticks show the closest and furthest distances observed from the snapshots.

interactions of residues T19, H20, N22, I23, D24, and W31 in the CDC37 N-terminal domain with residues Q451, Q454, G455, Y458, L459, H466, and V482-D486 in the RAF1 C-lobe, consistent with observations in a recent study[20]. It supports the notion that the CDC37 kinase recognition motif can be extended to [20]HPNID---SL—W[31], as described by Bjorklund et al. [26]. Moreover, we observe electrostatic interactions of RAF1 residues K462, R467 (HRD motif), and D486 (DFG motif) with CDC37 N-terminal domain residues E18, E38, and R30, respectively. The HRD and DFG motifs are implicated in RAF kinase activation and catalysis.

The most conserved residues of CDC37 homologs are in the N-terminal region (residues 1–30)[27,28], and comprise two of the three notable interaction interfaces observed between CDC37 and HSP90 (Supplementary Fig. 11a). The first interface involves contacts formed by CDC37 residues 1–10 (Supplementary Fig. 11b). CDC37 residues Y4 and W7 interact with residues L387 and Q396, located on the catalytic loop (residues 383–396) of HSP90-A. The catalytic loop of HSP90 contains the conserved R391 that interacts with the ATP γ-phosphate, thereby influencing its ATPase activity[29]. When CDC37 residues Y4 and W7 were mutated to

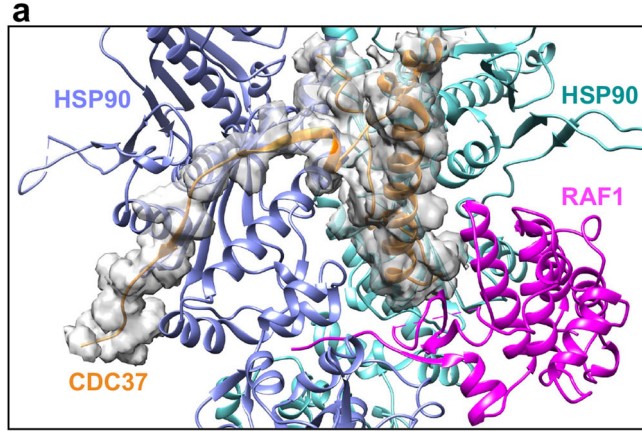

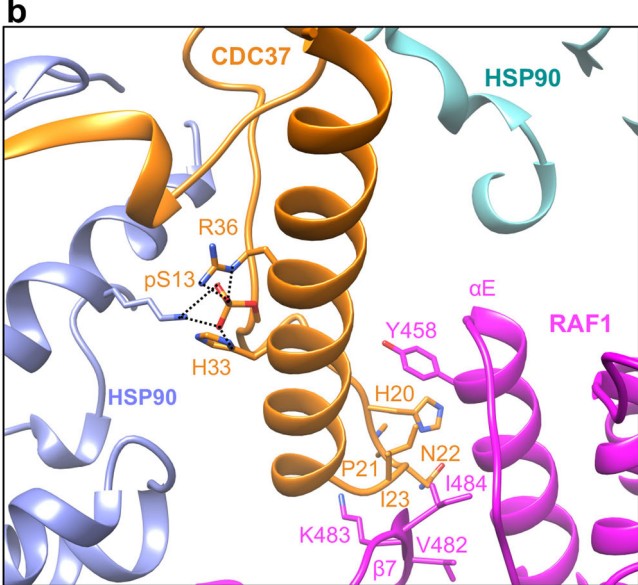

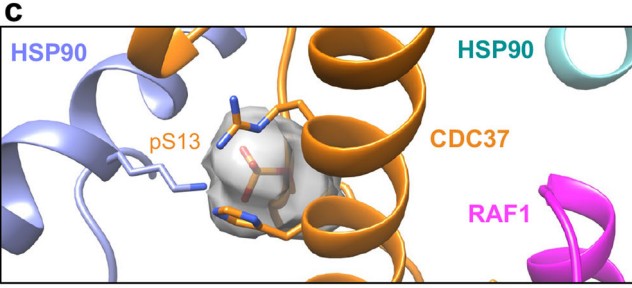

**Fig. 5 | The interactions formed by CDC37 in the RAF1-HSP90-CDC37 complex. a** An enlarged view of the structure of the RAF1-HSP90-CDC37 complex, showing the cryo-EM Coulomb potential for CDC37 in semi-transparent surface representation. **b** The conserved HxN motif of CDC37 mimics the interaction pattern of N-lobe residues with the C-lobe of RAF1. The phosphorylated CDC37 residue S13 interacts with R36 and H33 from CDC37 and with K405 from HSP90-B. These interactions are represented using dashed lines. **c** The cryo-EM Coulomb potential of the phosphorylated CDC37 residue S13 is displayed at a threshold of 0.02 in semi-transparent surface representation.

alanine in a different study, they were found to abolish client kinase binding activity completely[30]. The first interface also includes CDC37 residues D8 and H9 interacting with K36 and D188 of HSP90-B. At the second interface, CDC37 residues 10–18 form an extensive network of ionic interactions with two helical segments (residues Q396 to R404, Q445, and K446) of HSP90-B (Supplementary Fig. 11c). The third interface involves a β-strand on CDC37 (residues 120–129) that packs against β-strand (residues 317–323), which is

part of a β-sheet in the MD of HSP90-B (Supplementary Fig. 11d). Unlike the structure reported by García-Alonso et al.[20], the CDC37 MD is not resolved in our structure. This discrepancy may result from differences in our expression constructs, data collection, and processing strategies, potentially contributing to observed structural variations, including the absence of the CDC37 MD.

CDC37 phosphorylation at the conserved S13 site is a prerequisite for efficient binding to client kinases, and is crucial for recruiting the client kinase-CDC37 complex to HSP90[27,30,31]. In our structure, there is density supporting the assignment of phosphoserine13 (pS13), a structural element of CDC37. This residue forms salt bridge interactions with CDC37 residues R36 and H33 (Fig. 5c) that help to stabilize the CDC37 N-terminus coiled-coil, consistent with a recent study[20]. The electrostatic interaction between the phosphate (pS13) and HSP90-B K405 is the strongest pairwise-residue coulombic interaction of the complex (−171.11 kcal/mol). This interaction is preserved in the CDK4-HSP90-CDC37 structure. Taken together, our observation of pS13 (CDC37)-K405 (HSP90) and H20 (CDC37)-Y458 (RAF1) interactions, marked by the strongest coulombic and vdW forces in the complex, points to the essential role played by CDC37 in enabling HSP90-client interactions.

**Structures of HSP90 dimers in the semi-open state, and comparison with the closed state**

The semi-open state structure revealed HSP90 dimers, with only the NTD and MD visible in both protomers. No density associated with the CTD was observed, likely due to its flexibility or interactions with the air-water interface[32] (Fig. 6a, b). Unlike the closed state structure, RAF1 and CDC37 are unresolvable in this map. To mitigate the heterogeneity and further stabilize the conformational states present in the sample, we performed crosslinking with DSSO, and collected an additional cryo-EM dataset (Supplementary Figs. 8 and 9). The crosslinked semi-open map exhibited a global resolution of 3.9 Å, marginally lower than the non-crosslinked state. Nevertheless, it contains well-defined features more consistent with this resolution (Fig. 6c). Like the non-crosslinked semi-open structure (Fig. 6d), the complex demonstrated HSP90 dimerization, stabilized by the NTD and MD, while lacking the CTD (HSP90), RAF1, and CDC37 (Fig. 6e).

In the closed state, the nucleotide resides in the NTD of both HSP90 protomers (Supplementary Fig. 12a), surrounded by MD residues. R391 forms stabilizing interactions with the γ-phosphate of the nucleotide (Supplementary Fig. 12b, c), likely ADP-molybdate but modeled as ATP, as in most other structures of HSP90-CDC37-client protein complexes[19–21]. Molybdate is a phosphate analog, and molybdate with ADP has been suggested to mimic ATP or a transition state analog after ATP hydrolysis, with molybdate occupying the γ-phosphate site[33]. The presence of ADP-molybdate helped trap the closed conformation by stabilizing the RAF1-HSP90-CDC37 complex. Similarly, the semi-open state map revealed HSP90 dimers formed by NTDs bound to ADP-molybdate (Supplementary Fig. 12d–f). The semi-open state of the HSP90 dimer likely occurs after the dissociation of CDC37 and RAF1 but before HSP90 returns to an open state through structural changes triggered by ATP hydrolysis. This semi-open state structure exhibits a dimerization interface resembling the closed state, involving the NTD stabilized by ADP-molybdate and the MD residues, including the src-loops.

We performed 1 μs of atomistic simulation on the semi-open state complex, following a similar approach to the closed state. However, unlike the closed state, the HSP90 protomers in the semi-open state exhibit larger fluctuations, except for the stable core region of the dimer interface (Fig. 6f). To enable a direct comparison of structural fluctuations, we normalized the Root Mean Square Fluctuation (RMSF)-based B-factors for the semi-open, cross-linked semi-open, and closed states, as depicted in Figs. 6f, g and 1e. The RMSF-based B-factor visualization of the cross-linked semi-open model is shown in Fig. 6g, and it appears to fluctuate more than the closed and non-crosslinked semi-open state during the simulations. The likely reason for the larger fluctuations in the crosslinked semi-open structure, compared to the other two structures described in this study (Figs. 1e, 6f), is

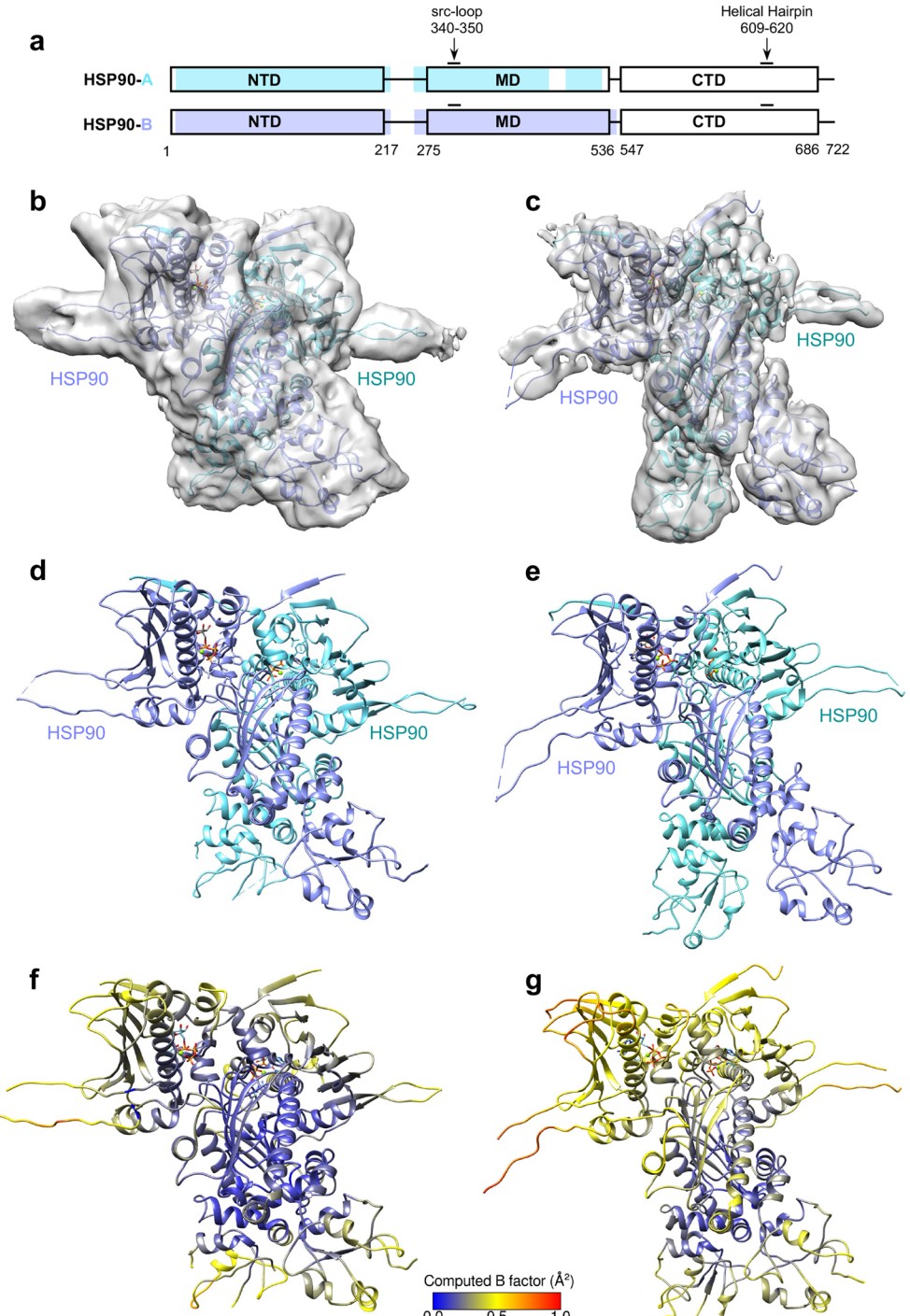

**Fig. 6 | Structures of the HSP90 dimer in the semi-open state. a** Domain architecture of the native HSP90 protomers A (cyan) and B (blue), showing regions observed in the map. The density for both protomers A and B in the map of the crosslinked HSP90 sample is resolvable for the same residue range observed in native protomer B (blue). **b, c** The overall structure of the HSP90 dimer in the semi-open state showing the electron potential map of the model obtained using the (**b**) native and (**c**) cross-linked samples. **d, e** The overall structure of the HSP90 dimer in the semi-open state shown in the cartoon representation obtained using the (**d**) native and (**e**) cross-linked samples. **f, g** The overall structure of the semi-open states, obtained using the (**f**) native and (**g**) cross-linked samples, colored by RMSF-based B-factor obtained from molecular simulation. The computed B-factor is normalized such that a direct comparison can be made to the closed state in Fig. 1e. Note that the increased fluctuation observed in the cross-linked structure, as compared to the native structures, is likely due to the absence of modeled DSSO crosslinks in our structure and simulations.

the presence of DSSO in our sample, but the absence of DSSO crosslinks in our structure and simulations. Although the crosslinked semi-open state sample used DSSO as a crosslinker, joining two nearby lysine residues each, we could not model DSSO molecules into our structure because they were not resolved in the cryo-EM maps. Crosslinking restricted the structure by holding lysine residues in proximity, and modified the charge of the

residues. When performing simulations with this structure lacking the crosslinks, we still observed the energetic consequences of the crosslink covalent restraints impacting the motions of the residues. Therefore, the higher observed residue fluctuations in our calculations could be attributed to the simulations being performed without DSSO. Our simulations further reveal that the HSP90 NTD β-strand swap upon ATP binding contributes

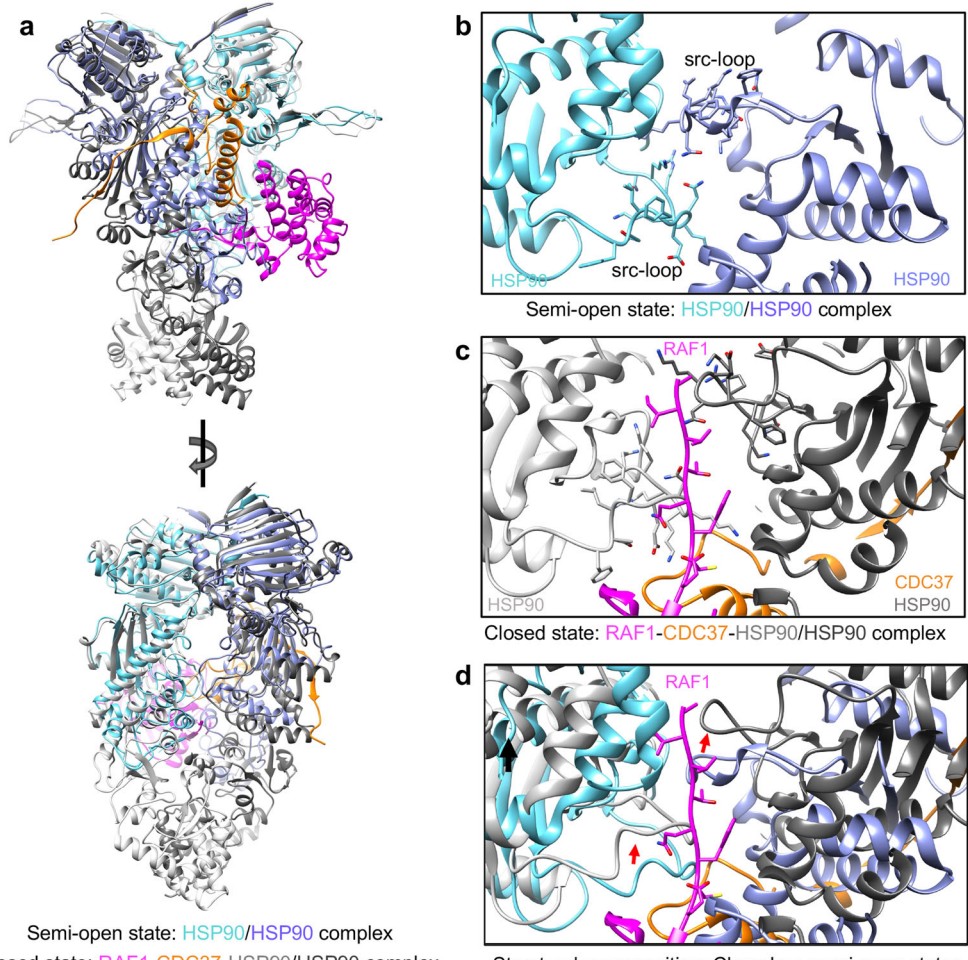

**Fig. 7 | Structural comparison between the closed and the semi-open state.**
**a** Structural superposition of the closed and semi-open (non-crosslinked sample) states aligned using the HSP90-A. The semi-open HSP90 promoters A and B are colored cyan and blue, respectively, while the closed-state HSP90 promoters A and B are colored light and dark gray, respectively. RAF1 and CDC37 in the closed state are shown in magenta and orange, respectively. **b** Enlarged view of the semi-open state showing the interactions formed by the residues in the src-loops of HSP90 promoters. **c** Enlarged view of the closed state, showing the interactions formed by the residues in the src-loops of HSP90 protomers (forming the luminal cavity) with RAF1 residues. The HSP90 protomers A and B are colored light and dark gray. **d** Structural superposition of closed and semi-open states shown in panels (**b**) and (**c**) highlighting the conformational changes (red arrows) in the src-loops between these two states.

substantially to the stability of the HSP90 dimer in both the closed and semi-open states (Supplementary Fig. 13) and represents an important HSP90 structural element. In the closed state, F15 (HSP90-A) interacts with F165 (HSP90-B) with a binding energy of −3.31 kcal/mol and a reciprocal interaction of −3.20 kcal/mol, and F17 (HSP90-B) also interacts favorably with F165 (HSP90-A), with a binding energy of −2.94 kcal/mol. Similarly, in the semi-open state, F15 and F17 (HSP90-B) interact with F165 (HSP90-A), forming two of the strongest stabilizing vdW interactions (−3.29 kcal/mol and −3.14 kcal/mol, respectively) in the HSP90 NTD, where the β-strand swap occurs. The energetic similarity of these HSP90 NTD interactions in both the closed and semi-open states further corroborates that the semi-open state remains stabilized by the NTD and suggests that the semi-open state likely exists between the closed and fully open states.

The semi-open state HSP90 dimer structure exhibits a Cα RMSD of 3.7 Å when compared to the closed state HSP90 dimer (Fig. 7a). Notably, a comparison of the src-loops from the closed and semi-open states reveals a shift in the relative position of the src-loops from both protomers (Fig. 7b, c). Unlike the closed state, the semi-open state lacks the luminal cavity containing the unfolded RAF1 β5-strand. In the semi-open state, the src-loop of each HSP90 protomer is directed toward the center of the dimer interface, contributing to dimer interface interactions (Fig. 7b). This conformational shift of the src-loop in the semi-open state prevents the formation of the luminal cavity observed in the closed state (Fig. 7d).

### Computational analysis of the closed and semi-open states reveals redistribution of molecular interactions

The Molecular Mechanics-Generalized Born Surface Area (MM-GBSA) predicts the binding free energies of protein-protein interactions[34,35]. We used MM-GBSA to show that the dimeric association between HSP90 protomers is the strongest protein-protein interaction within the closed-state complex (−276.72 kcal/mol). Additionally, the association of RAF1-CDC37 to the HSP90 dimer is the second strongest interaction (−241.89 kcal/mol). A comparison of closed and semi-open state energetics suggests that the HSP90 CTD strengthens interactions between the two HSP90 protomers (Supplementary Table 1). Indeed, the binding energy between HSP90 protomers for the crosslinked semi-open state (−195.49 kcal/mol) and the non-crosslinked semi-open state (−182.30 kcal/mol) are close in value (differing by 13.19 kcal/mol), while the closed state (−276.72 kcal/mol) is more favorable. Looking at the components of MM-GBSA energies, we see that the crosslinked semi-open state has less favorable coulombic energies compared to the non-crosslinked semi-open state and closed state (+532.38, +271.38, and −4.89 kcal/mol,

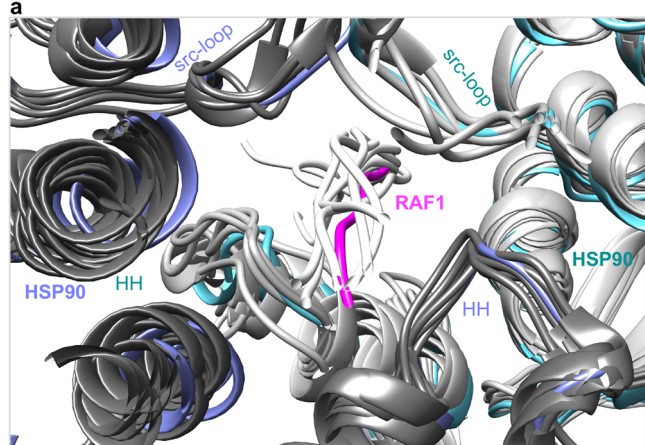

**Fig. 8 | Comparative structural analysis of the interaction between unfolded residues of the client proteins with the luminal cavity of the HSP90 dimer.**
**a** Enlarged view of the luminal cavity obtained using the structural superposition of the RAF1-HSP90-CDC37 complex with other structures of client proteins bound to HSP90 and cochaperone. **b** The table lists PDB IDs, residues present in the unfolded region of the client proteins, and the hydrophobicity and isoelectric point of these residues in all the structures used for comparative structural analysis in panel (**a**). The asterisk (*) denotes a discrepancy between the UniProt (P04049) and modeled sequences, where I419 and V420 are modeled as V419 and I420.

| PDB | Complex | Sequence | Hydrophobicity / PI |
|---|---|---|---|
| This study | RAF1-HSP90-CDC37 | 418-AIVTQWCEG-426 | 23.04/3.3 |
| 7Z38* | RAF1-HSP90-CDC37 | 418-AIVTQWCEG-426 | 23.04/3.3 |
| 7ZR0 | BRAF(V600E)-HSP90-CDC37 | 525-LAIVTQWCEG-534 | 28.00/3.3 |
| 5FWK | CDK4-HSP90-CDC37 | 88-KVTLVFEHVD-97 | 28.54/5.2 |
| 7KRJ | GR-HSP90-p23 | 524-TLPQLTPTLVS-534 | 30.48/6.0 |
| 7ZUB | AHR-HSP90-XAP2 | 276-SILEIRTKNF-285 | 28.34/10.1 |
| 8EOA | AIPL1-HSP90 | 1-FQGSHMDVS-9 | 16.16/4.9 |

X – Hydrophobic   X – Basic   X – Acidic   X – Polar

concentrated in the NTD. However, the largest movement in PC1 is observed in the HSP90-A MD. These fluctuations highlight the dynamic nature of the complex and semi-open dimeric state, and the movements of the molecular components comprising these states. Our computational approach augments structural observations made in this work by providing quantitative insight into the dynamic and energetic contributions driving the behavior of the closed and semi-open state complexes.

## Discussion

We determined the closed state structure of the RAF1-HSP90-CDC37 complex and the semi-open state structures of the HSP90 dimer and performed a detailed quantitative analysis of molecular interactions within these complexes using simulations. Our research, along with observations from the literature, offers context and an opportunity to examine three prevailing perspectives concerning the determinants of chaperone-mediated client folding. These three viewpoints are not necessarily mutually exclusive, and all may play roles in governing client-binding specificity.

Taipale et al. [36] suggested two fundamental principles governing client recognition: first, CDC37 recognizes a variety of protein folds, and second, the extent of interaction between HSP90 and the client is governed by the thermal and conformational stability of the client. The work focused on RAF family members ARAF, BRAF, and RAF1, due to their kinase domain sequence similarity (Supplementary Fig. 16), revealing differences in their association with HSP90. ARAF associated most strongly, followed by RAF1, while BRAF's binding was negligible. There is compelling evidence that HSP90 recognition is more closely related to the conformation or stability of the client rather than its sequence[37]. From the standpoint of client conformational stability being correlated to structural disorder, we observe disordered residues in the region of the RAF1 KD (residues 480-505) that interact with CDC37 in our closed state structure, whereas this region of BRAF, a 'weak' HSP90 client, remains mostly ordered. Our observation is consistent with recent work[20] that also identified disorder in RAF1 KD residues 480–505, and noted similar structural disorder in HSP90 client BRAF$_{V600E}$, which associates more strongly with HSP90 than wild-type BRAF[38–40]. Indeed, similarities are observed in a structural overlay of this RAF1 region (residues 480–505) from our work and from recent structures of RAF1[20] and BRAF$_{V600E}$[21]. However, the oncogenic activity of BRAF mutants does not necessarily correlate with their HSP90 association. For example, BRAF mutants, G597R and G467V, exhibit reduced kinase activity despite their stronger association with HSP90[37]. This observation is aligned with the HSP90 chaperone machinery's role in aiding the folding and maturation of unstable mutated oncoprotein clients in cancer cells[41].

Another prevailing observation[18] posits that hydrophobic residues at positions $i$ and $i + 3$ in the unfolded segment of the client, complementing features of the luminal cavity, may represent a conserved mode of HSP90-client recognition. Several cryo-EM structures of HSP90 and its homologs bound to client proteins have been reported in the closed state[18–22,42–45]. These structures include human HSP90 and HSP90 homologs from *Cricetulus griseus* and *Mus musculus*. ADP and molybdate, or AMPPNP, and in some cases, crosslinkers like glutaraldehyde or BS3, were utilized to stabilize and capture the closed state. The clients that exhibit luminal insertion vary, ranging from kinases to receptors, and in one striking case, the co-chaperone itself[43] (Fig. 8a, b). Our comparative structural analysis of the residues from various clients that thread through the luminal cavity suggests that the unfolded segment has an amphipathic character and contains hydrophobic patches. However, the position of these hydrophobic patches can vary (Fig. 8b). Our analysis supports the notion that the cavity is accommodating and demonstrates structural plasticity. We consistently observe that the HSP90 HH and src-loops frame the luminal cavity in all structures of client protein bound to HSP90 and cochaperone reported so far (Fig. 8a). We generated an interaction diagram of our closed state structure to highlight interactions between the RAF1 unfolded residues and the residues comprising the luminal cavity (Supplementary Fig. 17). Furthermore, we have illustrated in Supplementary Fig. 18 a structural overlay

respectively), although the energetic contribution from solvation diminishes this difference.

Our per-residue energetic decomposition analysis suggests that the semi-open state is characterized by a redistribution of interactions between the two HSP90 protomers that are present between RAF1 and HSP90 in the closed state. Specifically, we observe that interactions between the src-loop (residues 342–349) and other HSP90 MD residues (364–366, 369–375, 434, 435, 438, 441, and 442) in the semi-open state are largely absent in the closed-state complex (Supplementary Fig. 14). While src-loop residues of both HSP90 protomers form favorable contacts with the unfolded region (418–423) of RAF1 in the luminal cavity in the closed state, only the src-loop of HSP90-A forms peripheral interactions with RAF1 residues 477–483 (Supplementary Fig. 15).

We calculated the RMSF-based B-factor (Figs. 1e, 6f, g) to understand the extent of conformational sampling during our simulations, and performed principal component analysis (PCA) to quantify and visualize fluctuations and variance within the simulations. The first principal component describes the largest uncorrelated motion in a simulation. The motions captured by the first principal component (PC1) in the closed and semi-open states are shown in Supplementary Movies 1 and 2. PC1 reveals asymmetric fluctuations in the closed state between HSP90 protomers: HSP90-A generally exhibits larger fluctuations than HSP90-B. However, the largest movement in PC1 is observed in the HSP90-B NTD. Fluctuations are also observed in the RAF1 C-lobe, and in CDC37. In the semi-open state, asymmetric fluctuations are also observed between the HSP90 protomers: HSP90-B exhibits larger fluctuations than HSP90-A, with fluctuations

of client proteins in the luminal cavity, depicting their side chains colored by physicochemical characteristics (hydrophobic, basic, acidic, polar) (Supplementary Fig. 18a), and their complementarity to the amphipathic character of the luminal cavity (Supplementary Fig. 18b, c).

A third hypothesis[21] suggests that client stability is related to specific molecular interactions of the kinase DFG motif with CDC37, facilitated by hydrophobic contacts that permit the activation segment to occupy an inactive conformation. To corroborate this hypothesis, the authors provide the example of BRAF$_{V600E}$, in which hydrophobic interactions that would ordinarily stabilize an inactive kinase conformation in wild-type BRAF are disrupted due to the mutation. This disruption is thought to prevent the interaction of kinase DFG D594 with CDC37 R30, allowing CDC37 W31 to displace kinase F595. This region is modeled slightly differently in our structure than other recently solved client-HSP90 cochaperone complexes, possibly due to structural heterogeneity and differences in the local resolution of residues 480-505 in our structure (quantified and observed in 3DVA).

Our elucidation of the semi-open state also addresses an area of active interest in the field regarding identifying molecular configurations of client-chaperone machinery. An open "V-shaped" conformation of HSP90 was previously discussed in the literature, but is less well-characterized[46–48]. Multiple crystal structures of TRAP1, a mitochondrial HSP90 isoform, have been solved with and without the CTD. The TRAP1 crystal structure from zebrafish (PDB: 4IVG) was obtained using the full-length construct and resulted in proteolysis of the CTD. Crystal structures of human TRAP1 were obtained using constructs without the CTD, as the full-length protein was of low crystallographic quality, and the constructs without CTD diffracted to higher resolution[49–51]. Many structural features and interactions present in our semi-open state of HSP90 are preserved in these TRAP1 structures (Supplementary Fig. 19a). For example, the TRAP1 structures are dimerized via the N-terminal domains, and the nucleotide-binding sites are occupied by non-hydrolysable ATP analog in both the TRAP1 structures and our semi-open state (Supplementary Fig. 19b). However, both src-loops are visible in only one of these TRAP1 structures (PDB: 5HPH), and in this structure, one src-loop from one protomer contacts the src-loop from the opposing protomer, as seen in our semi-open state (Supplementary Fig. 19c). The semi-open state structure of the HSP90 dimer exhibits a Cα RMSD of 2.7 Å compared to the TRAP1 dimer (PDB: 4IVG), which suggests that it is more similar to the aforementioned TRAP1 than it is to the closed state. We rationalize that this semi-open conformation could represent a state at the end of the HSP90-CDC37-mediated kinase maturation cycle, following the dissociation of folded client and CDC37. Structurally, we observe this state held together by a trapped N-terminal dimer resulting from the presence of ADP-molybdate, and do not attempt to assign a specific biological significance to this semi-open state. Our biochemical data suggest that our structure is not a product of proteolytic cleavage, as observed in other structural work[49]. In addition to the NTD, the HSP90 CTD has also been implicated in nucleotide binding, although the literature on this topic is scarce[52–54]. Intriguingly, there is evidence of a cryptic nucleotide CTD binding site, which is only exposed upon NTD nucleotide binding[54]. Additionally, the CTD site exhibits divergent nucleotide specificity compared to the NTD site[54].

Our semi-open state exhibits two prominent features. Firstly, the src-loops from both HSP90 protomers reposition towards the dimer interface, obstructing luminal cavity formation. Secondly, HSP90 dimerization in the semi-open state involves NTDs exclusively, without dimeric interactions from disordered CTDs. The CTDs of HSP90 inherently form dimers, and transient N-terminal dimerization directly correlates with the ATPase cycle[10], where ATP binding and hydrolysis induce conformational changes, leading to NTD dimerization and disassembly[55].

This work highlights how structural elements within HSP90 and CDC37 impact the stability of the RAF1-HSP90-CDC37 complex. In HSP90, these structural elements include the src-loops, HH region, and N-terminal β-strand swap in the closed and semi-open states. In CDC37, these structural elements include the HxN motif and pS13. By analyzing the

closed and semi-open states using molecular simulations and energetic decomposition analysis, we were able to characterize and quantify molecular interactions involving HSP90 structural elements. These results reveal a redistribution of interactions between the two states, with RAF1 exhibiting asymmetric interactions with HSP90. Furthermore, by examining the structure and dynamics of the closed and semi-open states and comparing them to structures of various client proteins in the closed state, we provide valuable insights into the role of HSP90 and CDC37 in the folding and maturation of client proteins, including RAF1. Our findings suggest the following: (1) The HSP90 lumen is large and amphipathic, suggesting that it can accommodate diverse sequences from client proteins, and may not be limited to those containing a particular sequence motif. Our simulation analysis shows many weak and transient interactions between client proteins and the HSP90 residues lining the luminal cavity. (2) The HSP90 src-loop and HH play a crucial role in shaping the luminal cavity, where they interact with the unfolded residues of the client. Additionally, we observed a reorganization of interactions within the luminal cavity, including the HSP90 src-loop, when comparing the closed and semi-open states. (3) In the presence of an ATP analog, the dimeric interactions between the NTDs of two HSP90 protomers enhance the stability of both the closed and semi-open states, as noted by previous studies. (4) Our computational analysis suggests a role of CDC37 in bridging interactions between RAF1 and HSP90. In the closed-state complex, we observe that the strongest pairwise coulombic and vdW interactions feature CDC37: a salt bridge interaction between CDC37 (pS13) and HSP90-B (K405), and a vdW contact between CDC37 (H20) and RAF1 (Y458). These findings provide new insights into how HSP90 and CDC37 facilitate the folding and maturation of client proteins.

## Materials and methods
### Construction of bacmids for the production of RAF1-HSP90-CDC37 complex

Applying the insights from Vaughan et al.[56] and our previous work[57] with the co-expression of multiple genes during baculovirus infection, we designed our expression approach to include the co-expression of human CDC37 and human 14-3-3 (ζ isoform) to achieve a properly folded structure of RAF1. Considering that HSP90 proteins are endogenously expressed in insect cells at high levels, we co-expressed human RAF1, CDC37, and 14-3-3 in insect cells. This ensured that the resulting protein complex would contain only homogeneous, endogenously expressed HSP90 proteins, rather than a mixture of human and insect HSP90 proteins, which could have occurred if we had included human HSP90 in our polycistronic expression vector. The insect HSP90 from *Trichoplusia ni.* is of similar length to human HSP90, sharing approximately 82% sequence identity and 90% similarity with the two human HSP90 isoforms (Supplementary Fig. 2). Additionally, we designed the RAF1 expression construct to include a TEV protease-cleavable upstream maltose binding protein (MBP) fusion to enhance solubility, and we co-infected with a second bacmid that encoded TEV protease to cleave the MBP fusion protein in vivo during the infection. Supplementary Fig. 1 shows a schematic of the strategy employed.

We engineered a bacmid to include CDC37 and 14-3-3, leaving the *att*Tn*7* transposition location available for inserting RAF1 or similar mutants in future experiments. We started by transforming the *E. coli* recombineering strain, SW106 (provided by Dr. Don Court at the National Cancer Institute[58]), with DE32 bacmid DNA (bMON14272, which has deletions of the chitinase and cathepsin genes). A linear CDC37 knock-in cassette was made by generating amplicons for the AcMNPV polyhedrin promoter, human CDC37, *cat-sacB* selection/counter-selection marker, and ~300 bp left and right homology arms to the end of the *ODV-e52* gene. These amplicons were assembled via Isothermal (Gibson) Assembly with 40 bp overlaps to each previous amplicon. This linear CDC37 cassette was transformed into electro-competent cells of the SW106- bMON14272 Δ*chitinase-cathepsin E. coli* strain (electrocompetent cells were created after inducing the lambda RED function). Following selection on LB agar with kanamycin and

chloramphenicol, the intermediate construct of the CDC37 insertion was confirmed by PCR across the entire region. Next, a marker removal cassette comprised of ~300 bp of homology to the end of CDC37 and the downstream ODV-e52 homology arm, was assembled by overlap PCR. Electrocompetent cells were prepared from the intermediate CDC37 insertion strain (electrocompetent cells of the strain were created after inducing the lambda RED function) and these were transformed with the marker removal cassette. Colonies were selected on LB, no salt, 6% sucrose, and kanamycin agar plates. To verify the final CDC37 construction, an amplicon was generated using primers that bound outside the insertion region, and the amplicon was sequenced to confirm the correct bacmid (named DE41).

A similar approach was used to create the 14-3-3 linear knock-in cassette comprised of the AcMNPV polyhedrin promoter, 14-3-3ζ (YWHAZ), cat-sacB selection/counter-selection marker, and ~300 bp left and right homology arms to the middle of the ODV-e56 gene. This cassette was inserted into DE41 followed by marker removal in a similar fashion as described above for the creation of the CDC37 intermediate. Upon sequence confirmation of the insertions, the bacmid was isolated from SW106 via alkaline lysis and transformed into DE96, a DH10Bac derivative strain with improved helper plasmid and Tn7R-blocked genomic attTn7 site[59], followed by selection on LB agar with kanamycin and tetracycline, generating the final DE105 bacmid. The full genotype of this bacmid is bMON14272 chi-cath: e52:plh>CDC37; e56:plh>14-3-3.

To introduce RAF1 into the DE105 (CDC37/14-3-3) bacmid, we utilized the Bac-to-Bac system (Thermo Fisher Scientific). The RAF1 open reading frame (residues 1-648) was PCR amplified from template plasmid R777-E213 (Addgene #70497). The 5' primer added a Gateway attB1 cloning site and a tobacco etch virus (TEV) protease site (ENLYFQ/G) upstream of the first methionine of RAF1. The 3' primer included a C-terminal His6 tag, stop codon, and Gateway attB2 cloning site. Gateway BP recombination (Thermo Fisher Scientific) was used to introduce the attB-flanked PCR product into a Gateway Entry clone. The insert from this Entry clone was transferred by Gateway LR recombination to a baculovirus expression vector, pDest-795, which is a modified version of pFastBac 1 (Thermo Fisher Scientific) with an N-terminal maltose-binding protein (MBP) fusion. The final clone, designated R717-X22-795, encodes a fusion protein of MBP-tev-RAF1(1-648)-His6 and was transposed into DE105 (CDC37/14-3-3) per the manufacturer's instructions. Final bacmid DNA was validated by PCR amplification of the transposition junctions and used to produce baculovirus.

To achieve in vivo TEV protease cleavage of the MBP-tev-RAF1 fusion protein, we designed a separate bacmid for the expression of TEV protease. Specifically, the TEV protease gene was synthesized by ATUM with E. coli codon optimization, flanking attB1/attB2 Gateway sites, and three mutations, L56V/S135G/S219V, to improve cleavage[60]. An entry clone was constructed from the synthesized plasmid via a reverse Gateway BP reaction. The insert was then shuttled by Gateway LR recombination into pDest 8 baculovirus expression vector (ThermoFisher Scientific), modified from pFastBac1 to contain the Gateway attR1/attR2 recombination cassette. The final clone, R911-X26-008, was transformed into DE95[59] to generate the TEV protease bacmid. Final bacmid DNA was validated by PCR amplification of the transposition junctions and used to produce baculovirus.

**Virus production and insect cell protein co-expression.** Baculoviruses encoding heterologous TEV protease or RAF1/CDC37/14-3-3 were produced individually by transfection of bacmid DNA into Sf9 cells adapted to serum-free suspension and grown in SF900-III SFM Media (Thermo Fisher, Waltham MA). Briefly, 100 mL of Sf9 cells were set at $1.5 \times 10^6$ cells/mL and transfected using DNA:Cellfectin II (Thermo Fisher Scientific, Waltham MA) lipid complex. The DNA:Cellfectin complex was prepared using 70 μl of prepared bacmid DNA and 250 μL of Cellfectin II, both diluted in 500 μL of SF900 III media and mixed together. The transfected culture was incubated for 120 h at 27 °C in 250 mL Optimum Growth Flasks (Thomson Instrument Company Oceanside, CA), shaking at 105 RPM on an Innova 44 shaker with a 2" orbit (New Brunswick Scientific, Edison NJ). After the harvest of the cell culture supernatant at 120 h post-transfection, the baculovirus titer was measured by the ViroCyt 2100 (Sartorius Stedim New Oxford, PA).

Protein expression was carried out in Tni-FNL cells isolated and adapted at the Frederick National Laboratory for Cancer Research. Protein was expressed in 1 L of Sf900 III SFM media (Thermo Fisher Scientific Waltham, MA), set at $7 \times 10^5$ cells/ml in a 2.8 L Optimum Growth Flask (Thomson Instrument Company Oceanside, CA) the day before infection, so that the cells would double to ~$1.5 \times 10^6$ cells/mL on the day of infection for large scale purification. Co-infection was performed with an equal multiplicity of infection (MOI) of 3 for baculoviruses expressing RAF1/CDC37/14-3-3 and the TEV protease. Infected cell culture was incubated for 72 h at 21 °C, shaking at 105 RPM on an Innova 44 shaker with a 2" orbit (New Brunswick Scientific, Edison NJ). Cells were harvested, and cell pellets were stored at −80 °C until purification.

### Purification of RAF1-HSP90-CDC37 complex

During the purification, among the co-expressed proteins, only RAF1 had a C-terminal 6-His tag, and the other proteins did not carry any purification tag. One-liter culture pellet of Tni-FNL insect cells, which were transiently co-expressing human full-length RAF1, CDC37, along with 14-3-3ζ and TEV protease, was harvested through centrifugation and subsequently re-suspended in lysis buffer A (20 mM HEPES, pH 7.3, 150 mM NaCl, 1 mM TCEP, 10% glycerol (w/v), 10 mM MgCl₂, 20 mM Na₂MoO₄) supplied with 1:100 protease inhibitor cocktail (protease inhibitor cocktail, SIGMA P8849). Cells were lysed mechanically using a microfluidizer (Microfluidics) and cleared by centrifugation at 4 °C at 30,000 RCF for 30 min. Cleared lysate was then filtered with a 0.45 μm PES syringe filter and passed through a HisTrap HP column (Cytiva) pre-equilibrated with buffer A. Fractions containing the RAF1-HSP90-CDC37 complex were eluted from the column with an imidazole gradient and pooled to be run on a HiTrap SP HP column (Cytiva). It was observed that the majority of the co-expressed 14-3-3ζ came in the flow-through of the HisTrap column and was not part of the RAF1-HSP90-CDC37 complex. The ion-exchange (SP) step was necessary to clean out most of the impurities from the complex, where the complex eluted in the flow-through of the SP column in buffer B (20 mM HEPES, pH 7.3, 75 mM NaCl, 1 mM TCEP, 10% glycerol (w/v), 10 mM MgCl₂, 20 mM Na₂MoO₄). To further clean out any weakly associated 14-3-3ζ proteins and other minor impurities from the complex, SP eluent was passed through a HiTrap Heparin HP column (Cytiva). The complex was eluted from the heparin column using a NaCl gradient and the eluent fractions were pooled and concentrated with a 30 kDa cutoff spin concentrator (Millipore) to load on a Superose6 analytical size-exclusion column (Cytiva). The complex eluted under a single peak at 14 mL in the final SEC buffer C (20 mM HEPES, pH 7.3, 150 mM NaCl, 1 mM TCEP, 5% glycerol (w/v), 10 mM MgCl₂, 20 mM Na₂MoO₄). The fractions containing all three components of the complex (human RAF1 and CDC37 complexed with insect HSP90) corresponding to the apex of the SEC peak, were pooled, concentrated to 0.9 mg/mL and flash-frozen in liquid nitrogen in 50 μL aliquots. Protein was stored in −80 C freezer and a fresh aliquot was thawed and used for each cryo-EM experiment. We observed the presence of 14-3-3ζ in the eluted fractions during the initial purification stages (IMAC and ion exchange). However, during the subsequent heparin column step, all of the 14-3-3ζ proteins separated from the rest of the proteins, resulting in its absence in the fraction loaded onto the analytical SEC. The elution profile from the size-exclusion column chromatography, indicating peak fractions that correspond to the RAF1-HSP90-CDC37 complex, is presented in Supplementary Fig. 3a. Supplementary Fig. 3b shows the SDS-PAGE analysis of the peak fractions, displaying bands related to HSP90, RAF1, and CDC37. The purified complex was also analyzed using mass photometry, revealing a calculated molecular weight of 286 kDa. This aligns with the expected combined molecular weight of two protomers from insect

HSP90 and one protomer each from human RAF1 and CDC37 within the complex (Supplementary Fig. 3c).

For the crosslinking reaction, we mixed 100 μL of RAF1-HSP90-CDC37 complex (final concentration of 2.5 μM) with 96 μL of buffer C (20 mM HEPES pH 7.4, 150 mM NaCl, 10 mM $MgCl_2$, 20 mM $MoO_4$, 5% glycerol, and 1 mM TCEP). We then added 4 μL of DSSO (50 mM stock) to initiate the reaction, which proceeded for 1 h at room temperature. The reaction was halted by adding 4 μL of 1 M TRIS at pH 7.5. After centrifuging the sample at 10,000 g for 10 min, we loaded it onto a pre-equilibrated analytical SEC in buffer C and analyzed the peak fractions using SDS-PAGE. The fractions containing the crosslinked product were concentrated to 0.24 mg/mL and subsequently used for Cryo-EM grids. We pooled the peak fractions containing the crosslinked complex for structural studies.

### Cryo-EM sample preparation and data acquisition

The purified RAF1-HSP90-CDC37 complex was initially screened utilizing negative stain standard protocols[61] with uranyl formate before proceeding to cryo-EM. We stabilized the RAF1-HSP90-CDC37 complex through the addition of molybdate, which was also included during the purification. For cryo-EM, the grids were prepared by vitrification using a Vitrobot Mark IV (Thermo Fisher Scientific) operated at 4 °C and 98% relative humidity. Specifically, 3 μL aliquots of purified RAF1-HSP90-CDC37 complex at 0.2 mg/mL were placed on a holey carbon grid (300 Cu mesh, Quantifoil, R1.2/1.3) that was glow discharged at 15 mA for 5 s using a PELCO easi-Glow Discharge Cleaning System (Ted Pella, Inc., Redding, CA). Grids were blotted for 4.5 s with a blot force of 0 and plunge-frozen in liquid ethane cooled by liquid nitrogen. Grids were subsequently transferred to an FEI Titan Krios (Thermo Fisher Scientific) electron microscope operating at 300 kV with a Gatan K2 direct electron detector (Gatan, Inc., Pleasanton, CA). Dose-fractionated movie stacks of 40 frames were collected in counting mode using SerialEM software[62] at a nominal magnification of 29,000x, a defocus range of −1.5 to −3.0 μm, and a pixel size of 0.858 Å. An image shift of 3 × 3 multi-shot per stage movement (one hole per movie, 9 holes per stage position) was applied with an exposure time of 3.7 s and a total dose of 50 $e^-/Å^2$. Three datasets were collected, resulting in a total of 14,681 micrographs. Exemplar cryo-EM micrographs and 2D classes obtained from this sample are shown in Supplementary Fig. 4a, b, respectively. The grid preparation and data acquisition parameters for the crosslinked sample were also the same, except for the sample concentration, which was modified to 0.24 mg/mL. In total, 4479 micrographs were obtained for the crosslinked sample.

### Cryo-EM data processing

Despite multiple rounds of sample and grid optimization, preferential particle orientation and conformational heterogeneity hampered the quality of reconstructions obtained from the initial dataset, consisting of 5275 micrographs. To overcome these challenges, we collected two more datasets comprising 3542 and 5864 micrographs (Supplementary Fig. 5). A total of 14,681 micrographs were subjected to whole frame drift correction in MotionCor2[63] followed by contrast transfer function (CTF) estimation in CTFFIND4.1[64], both implemented in RELION3.1.3[65]. A total of 1,877,778 particles were automatically picked using the general model with crYOLO[66] from motion-corrected micrographs. Output box files from crYOLO were imported in RELION and particles were extracted. Extracted particles were subjected to multiple rounds of 2D classification (Supplementary Fig. 5). A de novo initial model from 601,572 particles was generated and used for multiple rounds of 3D classification that resulted in 3 distinct classes. Class 3 was selected, as it had recognizable density for HSP90, RAF1 and CDC37, and was used for 3D refinement, yielding a 3.7 Å reconstruction (R1 in Supplementary Fig. 5) that demonstrated heterogeneity in all components and hampered model building efforts. Further 3D classification using an initial model generated from class 3, utilizing the original 601,572 particles, allowed further separation of heterogeneity. This classification resulted in two predominant classes that were composed of 35% of the data and resembled a semi-open state (class 3), and 32% that resembled a closed state (class 5).

The closed state (class 5) was subjected to 3D refinement and resulted in a 3.7 Å reconstruction that showed a more resolved map in the HSP90 regions (R2 in Supplementary Fig. 5), but efforts at building the RAF1 region were still challenging. For the density corresponding to the dynamic RAF1 region, the volume was isolated by performing volume segmentation in UCSF Chimera1.16[67]. A mask was then generated using relion_mask_create (extended 8 pixels, and a soft edge of 4 pixels) and used for subsequent signal subtraction. Local masked 3D classification was performed without particle alignment in RELION[65], and this resulted in a conformational state using 48% of the particles. The resultant STAR file was reverted to the original, un-subtracted particles, followed by global 3D refinement of class 2. Bayesian polishing and CTF refinement were used to obtain a final reconstruction of 3.7 Å, and postprocessed in RELION (R3 in Supplementary Fig. 5, and Supplementary Fig. 6). The final reconstruction demonstrates similar attributes to the initial reconstructions, with less heterogeneity in all components than the first reconstruction (R1 in Supplementary Fig. 5), and a more resolved RAF1 region than the second reconstruction R2 in Supplementary Fig. 5, despite similar global resolution estimates.

The semi-open state (class 3) was used to generate an initial model that was used for further 3D classification with the original 601,572 particles (Supplementary Fig. 5). The classification culminated in a predominant class (class 4) with 52% distribution. 3D refinement and subsequent Bayesian polishing and CTF correction were used to obtain a final reconstruction of 3.5 Å (Supplementary Fig. 7). Postprocessing was carried out in RELION, where the resolution values reported for all reconstructions are based on the gold standard Fourier shell correlation (FSC) curve at 0.143 criteria. The resulting particle set for the closed and semi-open state was then subjected to 3D variability analysis in cryoSPARC3.2[68].

For the crosslinked sample, a total of 4479 micrographs were subjected to whole frame drift correction in MotionCor2 followed by contrast transfer function (CTF) estimation in CTFFIND4.1, both implemented in RELION3.1.3[65]. A total of 579,790 particles were automatically picked using the general model with crYOLO[66] from motion-corrected micrographs. Output box files from crYOLO were imported in RELION, and particles were extracted. Extracted particles were subjected to multiple rounds of 2D classification. A de novo initial model was generated and used for multiple rounds of 3D classification that resulted in 6 distinct classes. Two predominant classes comprised 39% of the data (class 3) and resembled the semi-open state, and 17% (class 1) resembled the closed state (Supplementary Fig. 8).

The crosslinked closed state (class 1) and the semi-open state (class 3) were both subjected to 3D refinement, Bayesian polishing, and CTF correction, and resembled the closed and semi-open state produced from the non-crosslinked data set (Supplementary Fig. 5). The crosslinked closed state map had a global resolution of 4.0 Å, albeit with less density for RAF1 and CDC37 than the non-crosslinked dataset. Due to the less resolved regions of the map and the lower particle count, we did not proceed further with this state. The crosslinked semi-open state had a global resolution of 3.9 Å. Like the non-crosslinked dataset, postprocessing was carried out in RELION, where the resolution values reported are based on the FSC curve at 0.143.

### 3D Variability analysis of the closed and semi-open states

During early rounds of data processing, 3D variability analysis (3DVA) performed using cryoSPARC[69] revealed significant structural heterogeneity and suggested the presence of multiple conformational states within the dataset. Despite multiple attempts at signal subtraction and local refinement, the regions of our map corresponding to RAF1 and CDC37 exhibited large variations in flexibility. Further rounds of iterative processing involving 3D classification, 3D refinement, Bayesian polishing, CTF refinement, signal subtraction, and 3D focused classification (Supplementary Fig. 5) led to the eventual elucidation of the closed state, and a semi-open state consisting of an HSP90 dimeric complex. Our subsequent analysis of variability on these maps verified that the flexibility of both the closed (Supplementary Movie 3) and semi-open state (Supplementary Movie 4) maps were continuous, indicating that no further heterogeneity was resolvable within the

data. This observation validates the utility of our workflow in resolving the states presented in this work.

## Model building and analysis

The closed-state map was sharpened with RELION as well as PHE-NIX1.20.1 half-map-based local anisotropic sharpening[70]. The semi-open map was sharpened with RELION as well as DeepEMhancer0.13 using the wideTarget training model[71]. The local resolution was estimated with PHENIX. Phyre2[72] was used to generate a homology model of the insect HSP90 dimer and was fit into the PHENIX local based sharpened map with anisotropic correction with Chimera. The closed state structure of CDK4-HSP90-CDC37 (PDB: 5FWK) was superimposed to obtain the CDC37 segment, and the RAF1 portion was obtained using the crystal structure of the RAF1 kinase domain (PDB: 3OMV). The combined model of the HSP90 dimer, CDC37, and RAF1 was then subjected to flexible fitting with iMODFIT1.03[73]. For the semi-open state, the same Phyre2 model of the insect HSP90 dimer, separated into protomers without RAF1 and CDC37 was used to fit into the DeepEMhancer sharpened map with Chimera and was subsequently flexibly fitted using iMODFIT. Sections that did not have corresponding density were removed, including regions of the CTD. The model was then refined iteratively with COOT0.9 and PHENIX. The final model was sharpened with model-based sharpening using LocScale0.1[74] for further iterative refinement in COOT, PHENIX, and ISOLDE1.4[75]. Parameters defining the data collection and the quality of the final atomic models are given in Table 1. Maps of the final closed state and semi-open state models at different threshold cutoffs are shown in Supplementary Fig. 20, and 3DFSC information for the closed and semi-open state models are shown in Supplementary Figs. 6b, 7b, and 9b.

The crosslinked semi-open map was sharpened with RELION as well as DeepEMhancer0.13 using the wideTarget training model[71]. The local resolution was estimated with PHENIX. The non-crosslinked semi-open structure was used to dock in the crosslinked map, and iMODFIT was used to flexibly fit the model. The model was then refined iteratively with COOT0.9 and PHENIX. The final model was sharpened with model-based sharpening with LocScale0.1[74] for further iterative refinement in COOT, PHENIX, and ISOLDE1.4[75]. Additional refinement and validation statistics for all three structures are provided in Table 1.

For structural alignments and comparisons, we used the Needleman-Wunsch alignment algorithm with the BLOSUM-62 matrix for super-impositions, implemented in Chimera. The structural analysis was performed using PDB codes 3OMV, 4IVG, 7Z38, 7ZR0, 5FWK, 7KRJ, 7ZUB, and 8EOA. For RMSD calculations, we used SSM superpose in COOT, which aligns based on the secondary structure. An interaction diagram of the unfolded residues of RAF1 interacting with the two HSP90 protomers was generated using the software LigPlot+[76]. Hydrophobicity and isoelectric point of the sequences shown in Fig. 8b were calculated by the Thermo-Fisher peptide analysis tool: https://www.thermofisher.com/us/en/home/life-science/protein-biology/peptides-proteins/custom-peptide-synthesis-services/peptide-analyzing-tool.html. Sequence alignments were performed using ClustalW[77], and the images were generated using ESPript 3.0[78].

## Molecular dynamics simulations and analysis

Molecular Dynamics simulations were conducted with AMBER 18[79] and prepared and analyzed with AmberTools 19[80] or AmberTools 20[81]. AMBER's TLEAP program was used to prepare all proteins for the simulations: The protein was parameterized using Amber FF14SB[82], and the phosphorylated S13 was parameterized using phosaa14SB parameters. The protein systems were placed in a box of TIP3P water such that all atoms were at least 10 Å from the boundary of the box. The ATP was parameterized with Antechamber, and AM1-BCC charges were added. For the closed state, the size of the complete model was 1554 protein residues plus two ATP residues, solvated in a TIP3P periodic solvent box of ca. size $143.8 \times 127.2 \times 11.8$ Å$^3$ containing 57,397 waters. For the semi-open state, the size of the complete model was 963 protein residues plus two ATP residues, solvated in a TIP3P periodic solvent box of ca. size

$101.7 \times 119.5 \times 118.1$ Å$^3$ containing 41,646 waters. For the semi-open (crosslinked) state, the size of the complete model was 999 protein residues plus two ATP residues, solvated in a TIP3P periodic solvent box of ca. size $125.4 \times 131.5 \times 114.7$ Å$^3$ containing 55,627 waters. The module PMEMD.cuda[83] was used to carry out simulations on graphics processing units (P100 GPUs). The equilibration run consisted of 4 minimizations of up to 40,000 steps (with each run having progressively reduced restraints, starting with all heavy atoms restrained, then releasing the waters and just restraining the proteins' heavy atoms), followed by six 20-ps runs at constant volume where the temperature of the simulation was raised from 0 to 298.15 K. Langevin dynamics[84] were used to maintain the temperature of the simulation with a collision frequency of 2.0 ps$^{-1}$. Next, a constant pressure (NPT) run allowed the volume of the box to adjust for 10.4 ns to maintain 1 bar of pressure, with the heavy atoms of the backbone of the proteins restrained. Finally, constant-volume (NVT) simulations were performed for 10 ns, under the same conditions as the subsequent production simulations, with just the termini restrained. Production NVT simulations were performed for 250 ns. For both semi-open and closed states, equilibration and production were run with four replicas, totaling 1 μs of production simulations. Only terminal residue backbone heavy atoms were restrained with a 5 kcal·mol−1·Å−2 force constant during production. The Shake algorithm[85] was used with a 2-fs time step. Periodic boundary conditions were applied, and the particle mesh Ewald[86] method was used to calculate long-range electrostatics. See Supplementary Methods, section 2.1. Molecular dynamics for additional details. Simulations were analyzed with structural and energetic analysis (See Supplementary Methods, section 2.2. RMSF-based B-factor calculation, section 2.3. Principal component analysis calculation, section 2.4. MM-GBSA, and section 2.5. Per-residue decomposition). The MM-GBSA thermodynamic cycle and schematics are shown in Supplementary Figs. 21 and 22, respectively. MM-GBSA results are shown in Supplementary Table 1. See Supplementary Section 3 for a discussion of caveats and Supplementary Section 4 for some concluding remarks about our simulation contributions to the study.

## Pull-down experiment

40 μL of Streptavidin-coated magnetic beads (Pierce, Thermo cat. 88816) were aliquoted into 1.5 mL Eppendorf tubes and washed twice with 500 μL of water. The beads were then equilibrated with binding buffer A (20 mM HEPES, pH 7.3, 150 mM NaCl, 1 mM TCEP, 0.05% Tween 20). Avi-tagged biotinylated KRAS (10 μg) loaded with either GDP or GMPPNP was incubated with the beads for 30 min at room temperature on a rotatory shaker. Unbound RAS was removed from the beads, and the beads were washed twice with binding buffer A. The beads were then incubated with the RAF1-HSP90-CDC37 complex (10 μg) in binding buffer B (20 mM HEPES, pH 7.3, 150 mM NaCl, 1 mM TCEP, 5% glycerol (w/v), 10 mM MgCl$_2$, 20 mM Na$_2$MoO$_4$, 0.05% Tween 20) for 1 h at room temperature with gentle rotation on a rotatory shaker. The flow-through (unbound RAF1) was collected at the end of the incubation period, and the beads were washed twice with binding buffer B. The beads were finally resuspended in 20 μL of elution buffer (binding buffer B). Due to the strong affinity of the biotin-streptavidin interaction, elution was performed by boiling the resuspended bead solution. Loads, flow-through (unbound RAF1), and elutions were analyzed on SDS-PAGE (Supplementary Fig. 10). Control experiments where the RAF1-HSP90-CDC37 complex was incubated with the beads without any RAS protein were performed in the same way, except that only buffer was used instead of RAS protein in the initial immobilization step.

## Statistics and reproducibility

For simulation studies, we ran four replicas for each state (closed, semi-open, and semi-open cross-linked). For MM-GBSA analysis, we report the mean and standard deviation internal to each of the replicas (Supplementary Table 1). We also report the mean, standard deviation (of the means), and standard error of the mean across the four replicas (Supplementary Table 1). The heatmaps shown in the main and Supplementary Figs. show the mean values of residue-residue interactions across all four replicas.

**Reporting summary**

Further information on research design is available in the Nature Portfolio Reporting Summary linked to this article.

**Data availability**

The atomic coordinates and cryo-EM maps of the closed state RAF1-HSP90-CDC37, the semi-open state HSP90 dimer, and the crosslinked semi-open state HSP90 dimer have been deposited to the Protein Data Bank and are available under accession numbers 8U1L, 8U1M, and 8U1N, and to the Electron Microscopy Data Bank with accession numbers EMD-41816, EMD-41817, and EMD-41818, respectively. For computational analysis, the starting structures, scripts for preparing, generating, and analyzing molecular simulations, and the final snapshots for each replica are available at https://github.com/tbalius/teb_scripts_programs/for_HSP_paper. All other data are available from the authors upon request.

**Code availability**

All code for scripts and programs used in this paper is available on GitHub at https://github.com/tbalius/teb_scripts_programs/, and a release of this repository is available on Zenodo[87] at https://doi.org/10.5281/zenodo.10607150. Code to reproduce the findings of this study is also available from the authors upon request.

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

## Acknowledgements

We thank William Gillette, Matt Drew, and Kelly Snead for their help in preparing recombinant proteins. We acknowledge Thomas Edwards, Natalia De Val, Htet Khant, Weimin Wu, and Rebecca Dillard for their help with this project. The authors also acknowledge the use of the Frederick Research Computing Environment (FRCE). This project was funded in whole or in part with federal funds from the National Cancer Institute, NIH Contract 75N91019D00024. The content of this publication does not necessarily reflect the views or policies of the Department of Health and Human Services, and the mention of trade names, commercial products, or organizations does not imply endorsement by the US Government.

## Author contributions

L.I.F. performed negative stain, cryo-EM grid preparation, data processing, model building, and refinement. M.C. and T.E.B. conducted molecular dynamics simulations and analysis. G.G. initiated the project, developed the methodology, and performed protein purification, sample preparation, and pull-down assay. C.G. carried out the cloning. J.F., T.F. and R.Y. assisted with cryo-EM data processing, 3D variability, and structural analysis. D.E., D.V.N., F.M. and D.K.S. contributed to the experimental design. L.I.F., M.C., T.E.B. and D.K.S. wrote the manuscript with input from all authors.

## Funding

## Competing interests

All authors, except F.M., have no competing interests. F.M. is a consultant for Ideaya Biosciences, Kura Oncology, Leidos Biomedical Research, Pfizer, Daiichi Sankyo, Amgen, PMV Pharma, OPNA-IO, and Quanta Therapeutics, has received research grants from Boehringer-Ingelheim, and is a consultant for and cofounder of BridgeBio Pharma.
