## [Peer Review File · Communications Biology]

Reviewers' comments:

Reviewer #1 (Remarks to the Author):

RAF kinases play critical roles in the MAP kinase signaling pathway. The proper folding and maturation of RAF requires Hsp90 and the kinase-specific co-chaperone CDC37. Here, the authors report three cryoEM structures: RAF1-Hsp90-CDC37 in a closed conformation and of an Hsp90 dimer in the “semi-open” conformation without and with crosslinking. In addition, the authors performed 1 μ s molecular dynamics simulation to further analyze the 3D structures and protein-protein interactions.

Major point:

1. RAF1-HSP90-CDC37 complex consists of mixed-species components consisting of insect HSP90 and human RAF1 and CDC37. Contrasting the work by Garcia-Alonso et al (2022), the MD of CDC37 is missing in the current structure. Can the authors comment on the potentially weaker interaction between Hsp90 and CDC37MD which might have been the result of using insect Hsp90 instead of human Hsp90?
2. In the abstract, line 9-12 “This analysis shows...semi-open state transition.” This observation is apparent from the 3D structure even without MD simulation. It is unclear what MD simulation adds to the work.
3. “HSP90” and “HSP” are used interchangeably, e.g. HSP90 NTD and HSP-NTD (p4 line 12 &14). Please change HSP to HSP90 throughout the paper for consistency.
4. P13 line 2-4, “This HSP90 dimer structure likely represents a semi-open state, occurring after the dissociation of CDC37 and RAF1, but before HSP90 returns to a fully open state through structural changes triggered by ATP hydrolysis.” This statement is unclear. Do the authors imply that this conformational transition is part of the HSP90 ATPase cycle? I understood that the semi-open state occurs after CDC37 and RAF1 release but with ATP still bound since ADP/molybdate mimicks the ATP transition state, or did the authors imply an ADP-bound state, or perhaps a conformational state after ADP release?
5. It would be helpful to present a more detailed structural analysis of the interaction between unfolded residues of the client proteins and the luminal cavity of the HSP90 dimer in Fig 8. Structural complementarity between unfolded residues and the surrounding lumen should be shown in terms of hydrophobicity, electrostatic properties, etc.

Minor point

1. P3 line 10. Please define “14-3-3”.
2. P3 lines 22-25. “HSP90 acts as...CDC37” should be removed. It is repeated in the following sentence.
3. Fig 1a. Mark the Src-loop and helical hairpin and label with residue numbers.
4. P5 line 9. “GR binding domains” should be “Glucocorticoid binding domains”
5. P7: Note R1, R2, R3 in the text as shown in Fig S5.
6. Fig 2a and Fig 5a legend. “showing the electron potential for CDC37 in surface representation”. However, the figures do not show the electron potential.
7. It would be better to show the actual cryoEM mass density in Fig. 2a and the electron potential as

surface representation in Fig. 5a.

8. P12 line 15. Should be Supplementary Fig. 12d.

9. P12 line 18-19. "there is apparent density that supports the assignment of phosphoserine 13". Please show the density in Fig 5.

10. P18 line 27. "more similar to TRAP1 than...". Which TRAP1 structure are the authors referring to? What is the RMSD between the two structures?

Reviewer #2 (Remarks to the Author):

The manuscript is well written and presents the EM structures of a closed Hsp90-RAF1-Cdc37 complex and an open Hsp90 structure that lacks the C-terminal domains that are normally found dimerised in the full-length protein. Similar structures have already been published (for example, Sara García-Alonso et al), which is mentioned in the manuscript. This prior art is addressed by an attempt to do an in-depth analysis of the structures and interactions between the individual components of the complexes. Some ideas around mechanism of the Hsp90 cycle are also introduced. However, to warrant its publication I feel the manuscript requires a higher level of critical analysis for many of the claims it presents. I make suggestions below and these concerns must be addressed to warrant publication, otherwise I feel that the data presented has already been previously published and publication will not be warranted.

1. There is no deposition file in the manuscript that I could see. This needs to be included. Access to the pdb coordinate files and EM maps need to be deposited and full access available once the structure is published.

2. In the section titled 'The unfolded β 5-strand' of the results, can you please give the residue numbers for the flexible src-loops (MD) and helical hairpins (HH) (CTD) from each HSP90 protomer. I would like to see a better-defined description of the structures and residues that line the lumen. My analysis shows it is more extensive than the loops described. Some residues form helical regions, for example, seem to line the lumen. Can you please give a comprehensive list of these residues as a whole?

3. For the section where the peptide runs through the lumen. It is unclear to me how well the amino-acid residues are identified for these peptide sequences. Is the EM density sufficient to allow a precise identification and conformation of the side-chains? A clear statement on this is required. If the precise sequence and conformation of side-chains cannot be identified, then what does it say about the conclusion that the lumen is accommodating and demonstrates structural plasticity? For example, if the EM density was at high resolution, perhaps side chains may show strict positional placement in the lumen and a consensus motif might be more evident. In relation to this, are the bound kinases showing the same binding conformations in the lumen (ie. CRAF, CDK4 and BRAF), which might be expected as they share very common peptide sequences bound in the lumen. Could Figure 8 b, be improved, perhaps

showing a strict residue alignment based on conservation at least for the kinases, but also taking into account a superimposition of these peptides in situ in their bound lumen states? That is, do the conserved residues overlay in these structures?

For example, In Braf we have AIVTQWCEGS and for Cdk4 we have KVTLVFEHVDN. I would expect these peptides to align well in the structure as they share a conserved sequence: Is that the case and what is the confidence that this is the case?

RAF1: NLAIVTQWCEGS

Cdk4: KVTLVFEHVDN

Can you derive a motif from such an analysis and compare to other clients to provide more in-depth analysis on this binding? Obviously, this is all dependant on clear identification and conformation of side chains for the lumen-bound peptide sequences of the clients.

4. I am also concerned that in the AIPL1 structure, the mouse AIPL1 sequence appears to start at MDVSL (UniProt entry: Q924K1), so is the upstream sequence part of a tag, which appears to be part of the threading through the Hsp90 lumen? GR is also truncated at one end of the thread. Does this affect the threading and the precise binding mode? Similarly, we see truncation for the Raf1 kinase domain. Could such truncations alter the exact binding of the peptides that thread through Hsp90? This needs to be clarified?

5. How do the Hsp90 residues F343, E344, R611 interact with other clients, which is covered for RAF1, but not the other clients. Are these universal client interacting residues?

6. I have concerns that the open structure may be a proteolytic fragment held together by a trapped N-terminal dimeric state resulting from the presence of molybdate. From previous structural experiments, we know such an interaction and structure is possible, which the authors explicitly refer to. The authors present no means of validating that this is not the case (ie a proteolytic fragment)? This is important because the authors describe this structure as a late-stage structure in the Hsp90 cycle, which ultimately may not be the case. If you cannot prove that this is a part of the Hsp90 cycle or specifically related to any part of the cycle, it should not be presented as such, but reported as an observed fragment in the experiments.

Furthermore, what biochemical evidence is there for C-terminal domain disassociation? I believe the literature on this is scarce, but non-the-less it should be referenced.

7. The HxN motif was found to be actually a larger motif that interacts with BRAF (see: Recognition of BRAF by CDC37 and Re-Evaluation of the Activation Mechanism for the Class 2 BRAF-L597R Mutant.

Dennis M. Bjorklund, R. Marc L. Morgan, Jasmeen Oberoi, Katie L. I. M. Day, Panagiota A. Galliou and Chrisostomos Prodromou. *Biomolecules* 2022, 12, 905. <https://doi.org/10.3390/biom12070905>). It is not immediately apparent how that study relates to the observations here. Can some comparison be given, or at least highlight differences found with that study?

8. Similarly, can the authors comment on the interactions between T19, H20, N22, I23, D24, and W31 of the CDC37 N-terminal domain with residues Q451, Q454, G455, Y458, L459, H466, and V482-D486 of the RAF1 C-terminal domain and how it relates to observations found in the above mentioned reference (Recognition of BRAF by CDC37....).

9. How do the interactions between RAF1 residues K462, R467 (of the kinase HRD motif), and D486 (of the kinase DFG motif) with CDC37 N-terminal domain residues E18, E38, and R30, relate to the findings in the Recognition of BRAF by CDC37 paper stated above.

10. In the study 'Recognition of BRAF by CDC37...' the authors found that the C-terminal domain of Cdc37 is important for client protein binding, but that is not evident in your structure. Perhaps some comment needs to be made here?

11. You mention an interaction, kinase DFG D594 with CDC37 R30, but in the study of 'Recognition of BRAF by CDC37...' such an interaction was not evident. Again, a statement is required. Why is there a difference? Are we looking at different structures between free Cdc37-kinase complex and that when bound to Hsp90? Some clarification is required on this point.

Reviewer #3 (Remarks to the Author):

This manuscript describes the cryo-EM structure of the complex of the RAF1 kinase domain with the chaperone protein HSP90 and a fragment of its co-chaperone protein CDC37. This complex structure in the closed state is compared with semi-open structures of the HSP90 dimer. The comparative structural analysis is completed by molecular dynamics simulations. Overall, this is an interesting work that provided additional insights into the structure, dynamics, and function of HSP90 in complex with RAF1 and CDC37 that expand recent observations (refs 20-22). Although I am not an expertise in cryo-EM, it appears that the work is high quality overall and the manuscript is well written. The MD simulations are also conducted and analyzed with care, and have provided useful information that enriched interpretation of the low-resolution experimental structures. That said, I have a few concerns/questions with the MD simulations.

First, it is not clear why the cross-linked HSP90 dimer structure is much more dynamic than the native

dimer (I would expect the opposite). Second, the MM-GBSA description in SI is a bit confusing despite the technique being well established, although still error prone. Third, the authors acknowledge some caveats in their simulation design (restraints, missing residues, simulation lengths etc) but their justification remains unconvincing.

Despite these somewhat minor weaknesses (some of which can be addressed in a revision), the work overall is expansive and rigorous, and it represents an important progress toward a more complete understanding of signaling through the RAS/MAPK signaling pathway.

Point-by-point response to the Reviewers' comments

We appreciate the reviewers for their constructive feedback and valuable suggestions. We have diligently addressed all reviewers' concerns within the revised manuscript, resulting in substantial enhancements. In the subsequent sections, detailed point-by-point responses to reviewers' comments are presented in light blue. Additionally, modifications to the main text are highlighted in dark blue, facilitating easier identification of revisions.

The editorial team requested that we shorten the revised manuscript to approximately 5,000 words to comply with the journal's guidelines. We achieved this by moving some of the cryo-EM processing details from the Results to the Methods section and meticulously revising other sections to reduce word count while preserving both clarity and necessary detail.

Reviewer #1 (Remarks to the Author):

RAF kinases play critical roles in the MAP kinase signaling pathway. The proper folding and maturation of RAF requires Hsp90 and the kinase-specific co-chaperone CDC37. Here, the authors report three cryoEM structures: RAF1-Hsp90-CDC37 in a closed conformation and of an Hsp90 dimer in the “semi-open” conformation without and with crosslinking. In addition, the authors performed 1 μ s molecular dynamics simulation to further analyze the 3D structures and protein-protein interactions.

Major point:

1. RAF1-HSP90-CDC37 complex consists of mixed-species components consisting of insect HSP90 and human RAF1 and CDC37. Contrasting the work by Garcia-Alonso et al (2022), the MD of CDC37 is missing in the current structure. Can the authors comment on the potentially weaker interaction between Hsp90 and CDC37MD which might have been the result of using insect Hsp90 instead of human Hsp90?

We thank the reviewer for the query concerning the absence of CDC37 MD in our structure. Comparing our structure to the one reported by García-Alonso et al., differences exist in the expression constructs (insect vs. human HSP90), data collection (presence or absence of tilted data), and processing strategies, potentially contributing to observed disparities. Our analysis showed that 5 of the 18 residues in HSP90 that interact with CDC37 (based on a 5 Å cutoff) differ between insect and human HSP90 proteins. These distinctions might explain structural variations, such as the absence of the CDC37 middle domain in our structure. We incorporated the following sentences on page 9 in response to the reviewer's point. “Unlike the structure reported by García-Alonso et al., the CDC37 MD is not resolved in our structure. This discrepancy may result from differences in our expression constructs (insect vs. human HSP90), data collection, and processing strategies, potentially contributing to observed structural variations, including the absence of the CDC37 MD.”

2. In the abstract, line 9-12 “This analysis shows...semi-open state transition.” This observation is apparent from the 3D structure even without MD simulation. It is unclear what MD simulation adds to the work.

We thank the reviewer and appreciate the opportunity to address the reviewer's concern regarding the significance of the molecular dynamics simulations in our abstract and this study.

These sentences in the abstract aimed to delineate the primary observations resulting from our quantitative and structural analyses of the molecular dynamics simulations. Our approach involved analyzing molecular interactions and energetics through atomistic models from cryo-EM structural data. The molecular dynamics analysis was not focused on observing a "transition", but rather on quantifying interaction strengths in both states for comparative purposes. This molecular dynamics analysis provided confidence in discussing residue-residue interactions and binding energies among the interacting proteins.

To address the reviewer's concern, we have modified the last two sentences of the abstract as follows: "This quantitative analysis demonstrated that CDC37 bridges the HSP90-RAF1 interaction, RAF1 binds asymmetrically to HSP90, and that HSP90 structural elements engage RAF1's unfolded region. Additionally, interactions at the N- and C-terminals stabilize HSP90 dimers, and molecular interactions in HSP90 dimers rearrange between the closed and semi-open states".

3. "HSP90" and "HSP" are used interchangeably, e.g. HSP90 NTD and HSP-NTD (p4 line 12 &14). Please change HSP to HSP90 throughout the paper for consistency.

We thank the reviewer for bringing this to our attention. We have now changed all instances of HSP to HSP90 throughout the manuscript.

4. P13 line 2-4, "This HSP90 dimer structure likely represents a semi-open state, occurring after the dissociation of CDC37 and RAF1, but before HSP90 returns to a fully open state through structural changes triggered by ATP hydrolysis." This statement is unclear. Do the authors imply that this conformational transition is part of the HSP90 ATPase cycle? I understood that the semi-open state occurs after CDC37 and RAF1 release but with ATP still bound since ADP/molybdate mimics the ATP transition state, or did the authors imply an ADP-bound state, or perhaps a conformational state after ADP release?

We appreciate the opportunity to further elucidate our observations on the semi-open conformational state. Given its limited characterization in the literature, our study aims to integrate the semi-open state into the established model of the physiological HSP90 client maturation cycle. Consistent with the maturation cycle presented in our manuscript, we propose that in the semi-open state, both CDC37 co-chaperone and RAF1 client have dissociated, indicating ATP hydrolysis. However, the inclusion of excess molybdate ions in our sample buffer resulted in the formation of ADP-molybdate within the semi-open complex. Because ADP-molybdate mimics the post-hydrolysis transition state of ATP, we postulate that in our structures, the conformational transition of semi-open HSP90 to its fully open state might be impeded by the presence of ADP-molybdate.

5. It would be helpful to present a more detailed structural analysis of the interaction between unfolded residues of the client proteins and the luminal cavity of the HSP90 dimer in Fig 8. Structural complementarity between unfolded residues and the surrounding lumen should be shown in terms of hydrophobicity, electrostatic properties, etc.

We thank the reviewer for their request to analyze the interactions between the unfolded client residues and the HSP90 luminal cavity in greater detail. To address the reviewer's point, we have generated a new figure with three panels (Supplemental Fig. 22) that depicts an overlay of the

unfolded residues of all client proteins listed in Fig. 8b within the luminal cavity of the HSP90 dimer. In these panels, the side chains of the client proteins are shown in stick representation and colored according to their physicochemical characteristics (i.e., hydrophobic, acidic, basic, and polar). The panels show the HSP90 luminal cavity in either cartoon representation (Supplemental Fig. 22a), or in surface representation colored by hydrophobicity (Supplemental Figs. 22b, 22c). To further elaborate upon the structural analysis conducted for our structure, we have also generated an interaction diagram of the unfolded residues of RAF1 interacting with the luminal cavity formed by the association of two HSP90 protomers. This diagram complements our depiction of the hydrophobicity of the lumen (Fig 2b) and shows hydrophobic interactions (red dashed lines) as well as electrostatic interactions (green dashed lines) between the unfolded client and HSP90 residues.

Supplemental Figure 17. Luminal interactions between RAF1 unfolded residues and HSP90. The program LigPlot (*J. Chem. Inf. Model.*, **51**, 2778-2786) was used to generate the interactions. RAF1 unfolded luminal residues are shown in purple. Polar interacting HSP90 residues are shown in tan. HSP90 residues that make van der Waals contacts are shown in red semicircles. Labels indicate the residue name and number, as well as what chain they belong to. Chains A, B, and C correspond to HSP90 protomer A, HSP90 protomer B, and RAF1, respectively. The label colors (red, blue, green, and orange) correspond to the src-loop, Helical Hairpin, Helix 16, and Helix 21, respectively.

Hairpin, Helix 16, and Helix 21, respectively, on either HSP90 protomer, and RAF1 labels are colored magenta. HSP90 residues that are not part of the listed secondary structure elements are labeled in black.

We added the following sentence to the third paragraph of the discussion section:

“We generated an interaction diagram of our closed state structure to highlight interactions between the RAF1 unfolded residues and the residues comprising the luminal cavity (**Supplementary Fig. 17**).”

Please see our response to reviewer 2, point 2, for additional analysis.

Minor point:

1. P3 line 10. Please define “14-3-3”.

To define 14-3-3, we have added this detail at the first use of this term: “a family of scaffolding molecules that modulate the function of other proteins through phosphorylation-dependent mechanisms (Roskoski, 2010; Terrell & Morrison, 2019).”

2. P3 lines 22-25. “HSP90 acts as....CDC37” should be removed. It is repeated in the following sentence.

We thank the reviewer for pointing this out. The second instance of the repeated sentences has been removed.

3. Fig 1a. Mark the Src-loop and helical hairpin and label with residue numbers.

We thank the reviewer for the suggestion to aid the reader. We have modified the figure.

4. P5 line 9. “GR binding domains” should be “Glucocorticoid binding domains”

The editorial team requested that we shorten the revised manuscript to approximately 5,000 words to comply with the journal's guidelines, and because of that, we no longer have the sentence containing this specific text. However, we have added a relevant citation to the first sentence of the paragraph that contained this text: “Several earlier studies have improved our understanding of the intricacies of chaperone-client interactions (Verba & Agard, 2017; Noddings et al. 2022)”.

5. P7: Note R1, R2, R3 in the text as shown in Fig S5.

We thank the reviewer for pointing this out. References to R1, R2, and R3 have been incorporated into the text as indicated in Supplementary Fig. 5.

6. Fig 2a and Fig 5a legend. “showing the electron potential for CDC37 in surface representation”. However, the figures do not show the electron potential.

We have revised Fig. 2a and Fig. 5a in the main figures to show the cryo-EM Coulomb potential maps in surface representation for all the displayed components of the complex. We hope that this clarifies any potential misunderstanding of the interpretation of the figures.

7. It would be better to show the actual cryoEM mass density in Fig. 2a and the electron potential as surface representation in Fig. 5a.

We have taken into consideration the reviewer's comment and modified Fig. 2a and Fig. 5a accordingly. We have modified the language of the captions of Figs. 2a and Figs. 5a to indicate that the figures depict the Coulomb potential of the cryo-EM map. The figure caption for Fig. 2a now states: "Enlarged view of the closed-state complex, showing the cryo-EM Coulomb potential map of RAF1 in semi-transparent surface representation." The figure caption for Fig. 5a now states: "An enlarged view of the structure of the RAF1-HSP90-CDC37 complex, showing the cryo-EM Coulomb potential for CDC37 in semi-transparent surface representation".

8. P12 line 15. Should be Supplementary Fig. 12d.

We thank the reviewer for pointing out this typo. This has now been corrected.

9. P12 line 18-19. "there is apparent density that supports the assignment of phosphoserine 13". Please show the density in Fig 5.

We have considered the reviewer's suggestion and now display the cryo-EM map for phosphoserine 13 of CDC37 as a separate figure panel (Fig. 5c).

10. P18 line 27. "more similar to TRAP1 than...". Which TRAP1 structure are the authors referring to? What is the RMSD between the two structures?

We thank the reviewer for requesting clarification of the TRAP1 structural comparison. This sentence has been revised to clarify that we are referring to the TRAP1 structure whose PDB ID was provided earlier in the same sentence. The sentence in the text now reads as follows: "The semi-open state structure of the HSP90 dimer exhibits a C α RMSD of 2.7 Å compared to the TRAP1 dimer (PDB: 4IVG), which suggests that it is more similar to the aforementioned TRAP1 than it is to the closed state."

Reviewer #2 (Remarks to the Author):

The manuscript is well written and presents the EM structures of a closed Hsp90-RAF1-Cdc37 complex and an open Hsp90 structure that lacks the C-terminal domains that are normally found dimerized in the full-length protein. Similar structures have already been published (for example, Sara García-Alonso et al), which is mentioned in the manuscript. This prior art is addressed by an attempt to do an in-depth analysis of the structures and interactions between the individual components of the complexes. Some ideas around mechanism of the Hsp90 cycle are also introduced. However, to warrant its publication I feel the manuscript requires a higher level of critical analysis for many of the claims it presents. I make suggestions below and these concerns must be addressed to warrant publication, otherwise I feel that the data presented has already been previously published and publication will not be warranted.

We thank the reviewer for their thoughtful review of our manuscript. We appreciate their positive feedback on the writing quality and the presentation of the structures. We also understand their concerns regarding the novelty and critical analysis of our findings in the context of existing literature. We agree that similar structures have been reported previously, as acknowledged in

the manuscript. However, we believe our work offers unique contributions beyond simply presenting new structures:

- **Using atomistic simulations for in-depth analysis of the structures and interactions:** We go beyond simply presenting the structures by performing 1-microsecond atomistic simulations, followed by an energetic decomposition analysis. This illuminates the dynamic nature of this complex and enhances our understanding of the interfaces and interactions between the individual components of the complexes.
- **New insights into HSP90 dynamics:** In addition to the RAF1-HSP90-CDC37 complex structure in the closed state, we also determined cryo-EM structures of HSP90 dimers in the "semi-open state," which showcases HSP90 dimers with their N-terminal and middle domains facilitating dimerization. This conformation offers a complementary perspective on the HSP90 dynamics, further contributing to our understanding of this essential molecular machinery.
- **Comparative structural analysis of various clients in complex with HSP90 complexes:** Our comparative structural analysis using structures of multiple clients in complex with HSP90 complexes reported so far reveals that unfolded client residues threading through the luminal cavity display an amphipathic nature with hydrophobic patches. However, the positions of these patches can vary. This analysis supports the idea of a highly accommodating cavity with notable structural plasticity.

In summary, the cryo-EM structures, combined with extensive simulations and energetic analyses presented in our manuscript, illuminate the dynamic nature of this complex and enhance our understanding of how these molecular interactions contribute to proper folding and function in this critical cellular signaling cascade.

We sincerely appreciate the reviewer's constructive feedback and have diligently addressed their suggestions for enhanced critical analysis. We believe these revisions significantly strengthen the manuscript's distinctiveness and contribution to the field.

1. There is no deposition file in the manuscript that I could see. This needs to be included. Access to the pdb coordinate files and EM maps need to be deposited and full access available once the structure is published.

We thank the reviewer for pointing out the missing information on accessing the structural data. The information has been incorporated into the "Data availability" section of the manuscript. Upon publication, the deposited coordinate file and EM maps will be fully accessible through the PDB and EMDB.

Here is the data availability section now added to the manuscript:

"Data Availability

The atomic coordinates and cryo-EM maps of the closed state RAF1-HSP90-CDC37, the semi-open state HSP90 dimer, and the crosslinked semi-open state HSP90 dimer have been deposited to the Protein Data Bank and are available under accession numbers 8U1L, 8U1M, and 8U1N, and to the Electron Microscopy Data Bank with accession numbers EMD-41816, EMD-41817, and EMD41818, respectively. For computational analysis, the starting structures, scripts for preparing, generating, and analyzing molecular simulations, and the final snapshots for each

replica are available at https://github.com/tbalius/teb_scripts_programs_for_HSP_paper. All other data are available from the authors upon request.”

2. In the section titled ‘The unfolded β 5-strand ...’ of the results, can you please give the residue numbers for the flexible src-loops (MD) and helical hairpins (HH) (CTD) from each HSP90 protomer. I would like to see a better-defined description of the structures and residues that line the lumen. My analysis shows it is more extensive than the loops described. Some residues form helical regions, for example, seem to line the lumen. Can you please give a comprehensive list of these residues as a whole?

We thank the reviewer for the question and the opportunity to elaborate. The src-loops are comprised of residues 340-350 of the MD, and the helical hairpins are comprised of residues 609-620 of the CTD from each HSP90 monomer. To obtain a more comprehensive description of the lumen, we have tabulated, in the **table below**, all residues within a 5.5 Angstrom threshold of the unfolded RAF1 residues 418-426, in accordance with our procedure for determining the luminal volume (see **Supplemental Methods**). We extended this analysis to include human HSP90-RAF1-CDC37 (PDB: 7Z38) from García-Alonso et al in the **table below**. We do see a few residues from the helical regions that also line the lumen, so in the revised manuscript, we have added this info: “This luminal cavity is mainly formed from residues comprised of flexible src-loops (MD) and helical hairpins (HH) (CTD) from each HSP90 protomer, as well as residues from α 16 (residues 423-443) and α 21 (residues 516-524), both from the HSP90 MD.” Figure 4b shows the results from our residue decomposition analysis of RAF1 interactions with both HSP90 protomers that further expand upon our analysis. For more information, please also see our response to **Reviewer 1, point 5**, and the LigPlot figure (new **Supplementary Figure 17**).

PDB: 8U1L (Our closed state structure):

HSP90 (Chain A)	HSP90 (Chain B)
PHE 343.A	PHE 343.B
GLU 344.A	GLU 344.B
ASN 345.A	ASN 345.B
LEU 438.A	LEU 438.B
HIS 441.A	HIS 441.B
GLU 442.A	GLU 442.B
TYR 519.A	ILE 516.B
GLN 608.A	TYR 519.B
ALA 609.A	GLN 522.B
LEU 610.A	GLN 523.B
ARG 611.A	MET 601.B
	MET 605.B
	GLN 608.B
	ALA 609.B
	LEU 610.B
	ARG 611.B
	MET 619.B

PDB: 7Z38 (Garcia-Alonso et al. closed state structure):

HSP90 (Chain A)	HSP90 (Chain B)
PHE 344.A	PHE 344.B
GLU 345.A	GLU 345.B
ASN 346.A	ASN 346.B
GLU 372.A	LYS 347.B
LYS 438.A	LEU 439.B
LEU 439.A	HIS 442.B
HIS 442.A	GLU 443.B
GLU 443.A	ILE 517.B
ILE 517.A	TYR 520.B
TYR 520.A	MET 606.B
GLN 523.A	GLN 609.B
GLN 524.A	ALA 610.B
TRP 598.A	LEU 611.B
MET 606.A	ARG 612.B
ALA 610.A	
LEU 611.A	
ARG 612.A	
MET 620.A	

3. For the section where the peptide runs through the lumen. It is unclear to me how well the amino-acid residues are identified for these peptide sequences. Is the EM density sufficient to allow a precise identification and conformation of the side-chains? A clear statement on this is required. If the precise sequence and conformation of side-chains cannot be identified, then what does it say about the conclusion that the lumen is accommodating and demonstrates structural plasticity? For example, if the EM density was at high resolution, perhaps side chains may show strict positional placement in the lumen and a consensus motif might be more evident. In relation to this, are the bound kinases showing the same binding conformations in the lumen (ie. CRAF, CDK4 and BRAF), which might be expected as they share very common peptide sequences bound in the lumen. Could Figure 8 b, be improved, perhaps showing a strict residue alignment based on conservation at least for the kinases, but also taking into account a superimposition of these peptides in situ in their bound lumen states? That is, do the conserved residues overlay in these structures?

For example, In Braf we have AIVTQWCEGS and for Cdk4 we have KVTLVFEHVDN. I would expect these peptides to align well in the structure as they share a conserved sequence: Is that the case and what is the confidence that this is the case?

RAF1: NLAIVTQWCEGS

Cdk4: KVTLVFEHVDN

Can you derive a motif from such an analysis and compare to other clients to provide more in-

depth analysis on this binding? Obviously, this is all dependant on clear identification and conformation of side chains for the lumen-bound peptide sequences of the clients.

We thank the reviewer for the question. To address the concerns of the reviewer regarding the EM density and resolvability of the luminal residue side chains, we have performed a comparative Q-score analysis of all the available PDB structures that we listed in Fig. 8 of the manuscript. The Q-score, which is a metric of atomic resolvability within a cryo-EM map, can be used to quantify the confidence of the model and each residue in this luminal region. Q-score values range from -1.0 to +1.0: -1.0 indicates that density and structure are anticorrelated; 0.0 indicates no correlation; and +1.0 indicates perfect correlation or strong agreement between density and structure. Our analysis reveals that different structures exhibit differing levels of variability, both on average and for individual residues (see below for the detailed analysis). Furthermore, no trend was observed when comparing the Q-scores of the residues based on their physicochemical properties (i.e., whether they are polar or hydrophobic). Even when considering the two highest resolution structures thus far available (2.85 and 3.1 Angstrom resolution, respectively), despite the similar extended conformation of the unfolded client residues in the lumen, their spatial orientation is not consistent, and therefore, a strict positional placement of the side chains that would suggest a consensus motif is not evident. Thus, our in-depth analysis further supports the variability of the luminal region and strengthens the assertion that the lumen demonstrates structural plasticity. Although we can appreciate the reviewer's expectation that conserved sequences should be structurally positioned within the luminal pocket in a similar fashion, our analysis shows that this is not the case. Additionally, we find that even for luminal residues from highly conserved clients (e.g., CDK4, BRAF, and RAF1), sequence identity in the luminal region does not imply a strict positional placement of the side chains of these residues. For instance, we note that the bulky sidechain (F93) of the CDK4 client in the lumen is positioned opposite to W523 in both RAF1 structures (this study and the one reported by Garcia-Alonso et al.), whereas in the structure containing BRAF, the corresponding bulky residue (W531) is translated in a position such that the entire unfolded peptide is embedded less deeply into the luminal pocket.

We have listed below the Q-Score of PDBs analyzed in Figure 8. The first column is the residue name, the second column is the residue number, and the third column is the residue Q-Score:

1. **PDB:8U1L (This study: RAF1-HSP90-CDC37)**, Average Q-score: 0.334, Resolution: 3.7 Å, CHAIN C

Res name	Res num	Q-score
ALA	418	0.534002
ILE	419	0.555561
VAL	420	0.292055
THR	421	0.446958
GLN	422	0.283124
TRP	423	0.265073
CYS	424	0.233648
GLU	425	0.236399
GLY	426	0.162664

2. PDB: 5FWK (CDK4-HSP90-CDC37), Average Q-score:0.356, Resolution:3.9 Å, CHAIN K

LYS	88	0.165594
VAL	89	0.19233
THR	90	0.477389
LEU	91	0.498693
VAL	92	0.542741
PHE	93	0.420224
GLU	94	0.135583
HIS	95	0.428602
VAL	96	0.461089
ASP	97	0.237771

3. PDB:7KRJ (GR-HSP90-p23), Average Q-score:0.648, Resolution:2.5 Å, CHAIN D

THR	524	0.508929
LEU	525	0.599969
PRO	526	0.428731
GLN	527	0.538837
LEU	528	0.769889
THR	529	0.648805
PRO	530	0.715897
THR	531	0.701463
LEU	532	0.743337
VAL	533	0.727709
SER	534	0.742753

4. PDB:7Z38 (RAF1-HSP90-CDC37), Average Q-score:0.522, Resolution:3.16 Å, CHAIN C

ALA	418	0.521445
VAL	419	0.268698
ILE	420	0.340898
THR	421	0.606799
GLN	422	0.605547
TRP	423	0.54647
CYS	424	0.70356
GLU	425	0.457567
GLY	426	0.647906

5. PDB:7ZRO (BRAFF^{V600E}-HSP90-CDC37), Average Q-score:0.481, Resolution:3.4 Å, CHAIN D

LEU	525	0.748474
ALA	526	0.619718
ILE	527	0.631233
VAL	528	0.719433
THR	529	0.412036
GLN	530	0.125155

TRP	531	0.546204
CYS	532	0.407214
GLU	533	-0.070889
GLY	534	0.674522

6. PDB:7ZUB (AHR-HSP90-XAP2), Average Q-score:0.683, Resolution:2.85 Å, CHAIN D

SER	276	0.65605
ILE	277	0.576782
LEU	278	0.677093
GLU	279	0.815805
ILE	280	0.477187
ARG	281	0.715483
THR	282	0.665309
LYS	283	0.768361
ASN	284	0.738734
PHE	285	0.739732

7. PDB:8EOA (AIPL1-HSP90), Average Q-score: 0.418, Resolution:3.9 Å, CHAIN C

PHE	1	0.252221
GLN	2	0.411851
GLY	3	0.293399
SER	4	0.252208
HIS	5	0.265209
MET	6	0.489349
ASP	7	0.539417
VAL	8	0.666003
SER	9	0.592296

4. I am also concerned that in the AIPL1 structure, the mouse AIPL1 sequence appears to start at MDVSL (uniprot entry: Q924K1), so is the upstream sequence part of a tag, which appears to be part of the threading through the Hsp90 lumen? GR is also truncated at one end of the thread. Does this affect the threading and the precise binding mode? Similarly, we see truncation for the Raf1 kinase domain. Could such truncations alter the exact binding of the peptides that thread through Hsp90? This needs to be clarified?

We thank the reviewer for noticing these structural details regarding the binding of HSP90 clients. We acknowledge the reviewer's concern about the impact of 'sequence artifacts,' such as expression tags or the lack or presence of specific modeled residues, on the observed binding of clients in the lumen. To be clear, the reviewer's mention of 'truncations' pertains to unmodeled residues that were present in the original protein constructs used in the structural studies; these residues comprise either an expression tag or the original sequence. The structural absence of such residues in the model does not reflect their absence in the protein but instead suggests conformational heterogeneity or dynamic fluctuations, resulting in a poorly defined map that hinders confident modeling of this region. Nonetheless, we acknowledge the possibility that the

sequence artifacts present in different kinase-HSP90-cochaperone complexes may influence the observed binding in the structures. It is however important to consider that the ‘exact binding’ of the unfolded residues of the client to HSP90 is an area of active investigation and has been proposed to relate to the transient unfolding of client kinase residues that are recognized by an HSP90-bound CDC37 cochaperone undergoing dynamic fluctuations (Keramisanou et al, Sci Adv . 2022 Mar 18;8(11):eabm9294); this is considered the ‘loading state’ of the client. Thus, the impact of the precise sequence of the partially unfolded conformations visited by the client and captured by the closed HSP90 dimer is not well understood at this point and is further impacted by the adaptability/plasticity of the luminal binding cavity, which presents highly dynamic binding interfaces that are involved in client interactions. To understand this mechanism, a well-resolved structure of the loading state of the client would be required, yet such a structure has thus far proven elusive to capture through single-particle cryoEM. It is also worth noting that the CDC37 cochaperone is kinase-specific, i.e., in the case of GR, this cochaperone is not involved, and in the case of AIPL1, the client itself functions as a cochaperone.

5. How do the Hsp90 residues F343, E344, R611 interact with other clients, which is covered for RAF1, but not the other clients. Are these universal client interacting residues?

We thank the reviewer for their question about the specific interactions of F343, E344, and R611 with other clients. We have performed this structural analysis for the other clients by identifying residues that interact within 5 Å of the specified HSP90 residues, which we have listed below.

PDB code (complex description)	Client residue information ^a
7Z38 (RAF1-HSP90-CDC37):	client residues 416, 418, 420-422, 424-425
7ZR0(BRAF ^{V600E} -HSP90-CDC37):	client residues 527-532, 586
5FWK (CDK4-HSP90-CDC37):	client residues 87-91, 93, 94
7KRJ (GR-HSP90-p23):	client residues 524, 526, 527, 529, 530
7ZUB (AHR-HSP90-XAP2):	client residues 272-276, 278, 281-284, 303
8EOA (AIPL1-HSP90):	client residues 1, 3-7. It should be noted that residues 1, 3-5 are considered to be part of the His6/TEV-tag that precedes the native sequence.
^a . Client residues that are within 5 Å of HSP90 residues F343, E344, and R611	

The above analysis shows that these specific HSP90 residues (F343, E344, and R611) appear to interact with all of the clients for which structural data is available thus far; however, the nature of these interactions is not uniform across various clients.

6. I have concerns that the open structure may be a proteolytic fragment held together by a trapped N-terminal dimeric state resulting from the presence of molybdate. From previous structural experiments, we know such an interaction and structure is possible, which the authors explicitly refer to. The authors present no means of validating that this is not the case (ie a proteolytic fragment)? This is important because the authors describe this structure as a late-stage structure in the Hsp90 cycle, which ultimately may not be the case. If you cannot prove that this is a part of the Hsp90 cycle or specifically related to any part of the cycle, it should not be presented as such, but reported as an observed fragment in the experiments. Furthermore, what

biochemical evidence is there for C-terminal domain disassociation? I believe the literature on this is scarce, but non-the-less it should be referenced.

We thank the reviewer for bringing up this point of discussion. Based on the results obtained from size exclusion, SDS-PAGE, Western Blot, and Mass Photometry experiments included in Supplementary Figure 3, we do not see any indication of a 'proteolytic fragment.' In our cryoEM experiments, the purified protein was used to prepare the grid and promptly flash-frozen for data collection, minimizing the risk of proteolytic cleavage commonly linked to crystallization experiments involving extended incubation at ambient temperatures. Therefore, we respectfully disagree with the assertion that our observed semi-open state should be reported as an observed "proteolytic fragment." All previous examples of this state from known literature have been presented, as the reviewer aptly notes. Of the examples provided, one was from crystallography that provided evidence of proteolytic cleavage of the sample using SDS-PAGE. In the other crystallographic examples provided in the manuscript, the authors explicitly chose to utilize constructs without the C-terminal domains. In our discussion, we make sure to cautiously attribute the lack of the CTD to either interaction at the air-water interface (as discussed in Wang et al.) or flexibility. We do not attempt to provide unequivocal proof that our observed semi-open structures represent a specific part of the HSP90 cycle. Rather, we try to speculatively rationalize our observations based on current literature in the field. Regardless of the biological significance of this state, we believe that it is an intriguing observation and provides valuable structural insight.

To address the concerns of the reviewer, we have modified the language of the manuscript to more explicitly indicate that our objective was not to definitively claim the existence of the semi-open state as a late-stage structure in the HSP90 cycle but rather to guide the reader by rationalizing the implication of this structural observation in the context of the current understanding in the field. We have removed the text stating that the semi-open state is "prior to the structural rearrangements resulting in the dimeric HSP90 open state", as we acknowledge that this language can cause confusion when considering the HSP90 cycle. We have also added a few references that provide biochemical evidence for a nucleotide binding site on the C-terminal domain (CTD) of HSP90. Although the literature on this topic is limited, as noted by the reviewer, we think that the inclusion of these references will be helpful to readers as a topic for consideration and further exploration.

The modified text, added to the discussion section, now reads as follows: "We rationalize that this semi-open conformation could represent a state at the end of the HSP90-CDC37-mediated kinase maturation cycle, following the dissociation of folded client and CDC37. Structurally, we observe this state held together by a trapped N-terminal dimer resulting from the presence of ADP-molybdate, and do not attempt to assign a specific biological significance to this semi-open state. Our biochemical data suggest that our structure is not a product of proteolytic cleavage, as observed in other structural work (Lavery et al.). In addition to the NTD, the HSP90 CTD has also been implicated in nucleotide binding, although the literature on this topic is scarce (Garnier et al., Marcu et al., Soti et al.). Intriguingly, there is evidence of a cryptic nucleotide CTD binding site, which is only exposed upon NTD nucleotide (Soti et al.). Additionally, the CTD site exhibits divergent nucleotide specificity compared to the NTD site (Soti et al.)."

7. The HxN motif was found to be actually a larger motif that interacts with BRAF (see: Recognition of BRAF by CDC37 and Re-Evaluation of the Activation Mechanism for the Class 2 BRAF-L597R

Mutant. Dennis M. Bjorklund, R. Marc L. Morgan, Jasmeen Oberoi, Katie L. I. M. Day, Panagiota A. Galliou and Chrisostomos Prodromou. *Biomolecules* 2022, 12, 905. <https://doi.org/10.3390/biom12070905>). It is not immediately apparent how that study relates to the observations here. Can some comparison be given, or at least highlight differences found with that study?

We thank the reviewer for highlighting this relevant literature on the CDC37 kinase recognition motif in the context of BRAF. In this study, the authors notice that the HPNI motif of CDC37 can potentially be extended to include residues 23, 24, 27, 28, 31, and 34. In our manuscript, we have mentioned residues I23, D24, and W31, which result from our energetic decomposition analysis, as having favorable van der Waals contacts with the RAF1 client. We also identified from our structural analysis the interaction of S27 with RAF1. Our findings corroborate the suggestion in the paper that “the motif for the recognition of the C-terminal lobe of kinase domains could be extended to include ²⁰HPNID---SL-W³¹”. We have modified our sentence in the text to point out these similarities as follows: “Our energetic decomposition analysis further reveals van der Waals interactions of residues T19, H20, N22, I23, D24, and W31 in the CDC37 N-terminal domain with residues Q451, Q454, G455, Y458, L459, H466, and V482-D486 in the RAF1 C-lobe, consistent with observations in a recent study²⁰. It supports the notion that the kinase recognition motif of CDC37 can be extended to consist of ²⁰HPNID---SL—W³¹, as described by Bjorklund et al.”.

8. Similarly, can the authors comment on the interactions between T19, H20, N22, I23, D24, and W31 of the CDC37 N-terminal domain with residues Q451, Q454, G455, Y458, L459, H466, and V482-D486 of the RAF1 C-terminal domain and how it relates to observations found in the above mentioned reference (Recognition of BRAF by CDC37....).

We thank the reviewer for their helpful comment. As the reviewer has noted in their question, we have discussed the favorable van der Waals contacts of CDC37 residues T19, H20, N22, I23, D24, and W31 of the CDC37 N-terminal domain with residues Q451, Q454, G455, Y458, L459, H466, and V482-D486 of RAF1.

Based on the sequence alignment of these RAF1 residues with BRAF, the corresponding residues are Q559, Q562, G563, Y566, L567, H574, and V590-D594. In the referenced paper, we find that the only residue among these that appears to be discussed is D594, in the context of mutation D594V in the V600E background. It is possible that these differences may be a result of differences in BRAF versus RAF1, or a result of the fact that these experiments were performed in the BRAF V600E background. However, the paper does mention the importance of the CDC37 N-terminal domain (residues 8-41) as a component of the ‘minimal elements’ required for high affinity binding to kinases, when examined in the context of binding to sBRAF V600E. This residue range encompasses residues 19, 20, 22, 23, 24, and 31, which we have noted in our manuscript.

9. How do the interactions between RAF1 residues K462, R467 (of the kinase HRD motif), and D486 (of the kinase DFG motif) with CDC37 N-terminal domain residues E18, E38, and R30, relate to the findings in the Recognition of BRAF by CDC37 paper stated above.

We thank the reviewer for the helpful comment. As the reviewer has noted in their question, in our manuscript we have discussed the favorable electrostatic interactions of RAF1 residues K462, R467 (of the kinase HRD motif), and D486 (of the kinase DFG motif) with CDC37 N-terminal domain residues E18, E38, and R30, and mentioned the importance of the HRD and DFG motifs

in kinase activation and catalysis. Based on the sequence alignment of these RAF1 residues with BRAF, the corresponding residues are K570, R575, and D594. Among these three BRAF residues, only the D594V mutation has been discussed in Bjorklund et al., which resulted in impaired kinase activity with no impact on interaction with CDC37. In addition to the possible effects of the V600E mutation background and differences between BRAF and RAF1 that we have alluded to in our response to the prior question, other factors that may explain the differences in our observations include the transient nature of the client-cochaperone interactions as well as our use of insect HSP90 as opposed to human HSP90.

10. In the study 'Recognition of BRAF by CDC37...' the authors found that the C-terminal domain of Cdc37 is important for client protein binding, but that is not evident in your structure. Perhaps some comment needs to be made here?

We thank the reviewer for pointing out this observation. As evidenced by our cryo-EM workflow to obtain the final model, there was significant compositional heterogeneity in the closed state complex. We employed sophisticated processing strategies to clean up the particles, including multiple rounds of 3D classification and refinement, signal subtraction, and focused classification. After obtaining a suitable reconstruction that enabled us to confidently build all components, this data was further subjected to 3D variability analysis. Our analysis, as well as the analysis provided by García-Alonso et al., further corroborate that this is indeed a highly flexible complex. As we have shown in Figure 1 of the manuscript, neither the MD nor CTD of CDC37 was resolvable in our complex structure, likely due to higher flexibility in this complex. The absence of these domains does not necessarily indicate their lack of importance in client protein binding but may instead be a consequence of the inherent flexibility of CDC37 CTD and/or conditions under which the structural data was obtained in our work.

11. You mention an interaction, kinase DFG D594 with CDC37 R30, but in the study of 'Recognition of BRAF by CDC37...' such an interaction was not evident. Again, a statement is required. Why is there a difference? Are we looking at different structures between free Cdc37-kinase complex and that when bound to Hsp90? Some clarification is required on this point.

We thank the reviewer for their question about the interaction of the kinase DFG motif with CDC37. Considering the absence of structural data for this region in our structure, we cannot compare our structural data with the data reported by Oberoi et al.

It is possible that the presence of HSP90 in the structure obtained by Oberoi et al. results in induced molecular interactions between CDC37 and the client. This may not ordinarily occur when CDC37 associates with the client in solution prior to binding with HSP90. However, as discussed by Oberoi and colleagues, this effect of BRAF V600E in disrupting the BRAF D594 – CDC37 R30 interaction remains a hypothesis based on their structural observations, and we are unaware of other evidence that corroborates this hypothesis.

Reviewer #3 (Remarks to the Author):

This manuscript describes the cryo-EM structure of the complex of the RAF1 kinase domain with the chaperone protein HSP90 and a fragment of its co-chaperone protein CDC37. This complex structure in the closed state is compared with semi-open structures of the HSP90 dimer. The comparative structural analysis is completed by molecular dynamics simulations. Overall, this is

an interesting work that provided additional insights into the structure, dynamics, and function of HSP90 in complex with RAF1 and CDC37 that expand recent observations (refs 20-22). Although I am not an expert in cryo-EM, it appears that the work is high quality overall and the manuscript is well written. The MD simulations are also conducted and analyzed with care, and have provided useful information that enriched interpretation of the low-resolution experimental structures. That said, I have a few concerns/questions with the MD simulations.

First, it is not clear why the cross-linked HSP90 dimer structure is much more dynamic than the native dimer (I would expect the opposite).

We thank the reviewer for their question regarding the dynamics of the crosslinked semi-open HSP90 dimer structure. As the reviewer noted, we point out in the text and in Fig. 6g that the crosslinked structure exhibits higher fluctuations than the closed and non-crosslinked semi-open states. As mentioned in the manuscript, the crosslinked semi-open state sample was prepared using DSSO as a crosslinker, where each DSSO molecule likely conjoined two nearby lysines. However, we could not build DSSO into our structure as they were not resolved in the cryo-EM maps. Crosslinking by DSSO constrained the structure by holding lysine residues in proximity and altered the charge of the residues. When performing the simulation with this structure in which crosslinks were absent, we observed the energetic consequences induced by the crosslinker without having explicit restraints on the motions of the residues impacted by the DSSO molecules. The higher observed residue fluctuations in our calculations could thus potentially be attributed to the simulations being performed without DSSO.

To address the reviewer's concern, we have added the following text to the figure caption of Fig. 6g for clarity: "Note that the increased fluctuation observed in the cross-linked structure, as compared to the native structures, is likely due to the absence of modeled DSSO crosslinks in our structure and simulations."

We have also added the following sentences on pg. 10, after our statement about the larger RMSF of the cross-linked structure: "The likely reason for the larger fluctuations in the semi-open crosslinked structure, compared to the other two structures described in this study (Figs. 1e, 6f), is the presence of DSSO in our sample, but the absence of DSSO crosslinks in our structure and simulations. Although the crosslinked semi-open state sample used DSSO as a crosslinker, joining two nearby lysine residues each, we could not model DSSO molecules into our structure because they were not resolved in the cryo-EM maps. Crosslinking restricted the structure by holding lysine residues in proximity, and modified the charge of the residues. When performing simulations with this structure lacking the crosslinks, we still observed the energetic consequences of the crosslink covalent restraints impacting the motions of the residues. Therefore, the higher observed residue fluctuations in our calculations could be attributed to the simulations being performed without DSSO."

Second, the MM-GBSA description in SI is a bit confusing despite the technique being well established, although still error-prone.

We thank the reviewer for this feedback. In the revised manuscript, we have made the MM-GBSA description clearer by making the following changes:

We added the following line to the method section on page 25: The MM-GBSA thermodynamic cycle and schematics are shown in **Supplementary Figs. 20 and 21**, respectively.

We added two SI figures:

Supplementary Figure 20. Thermodynamic cycle cartoon to illustrate MM-GBSA calculation. Two proteins (colored blue and cyan) come together to form a complex in vacuum (top) and water (bottom).

Supplementary Figure 21. Schematic illustration showing estimated binding energies for four different complex formations. HSP90-A, HSP90-B, RAF1, and CDC37 are colored cyan and blue, magenta, and orange, respectively (same color scheme as in Figure 1). Binding energies were calculated by the MM-GBSA method for each replica simulation. **a.** For the semi-open states (both semi-open and semi-open-cross-linked), the binding of the two HSP90 protomers forming the dimer. The jagged edge indicates the missing C-terminal domain. **b.** For the closed state, the binding of HSP90 protomers forming the dimer. **c.** For the closed state, the

binding of RAF1 and CDC37 to HSP90 dimer. **d.** For the closed state, the binding of RAF1 to the CDC37-HSP90 dimer complex.

We also modified the text on SI pages 3-4:

“We calculate an approximation to binding energy for two species to form a complex illustrated in the thermodynamic cycle (**Supplementary Figure 20**).”

“By looking at the thermodynamic cycle and summing the arrows (**Supplementary Figure 20**), we obtain the following equivalency: “

“...and calculated five binding energies (**Supplementary Figure 21** and **Supplementary Table 1**).”

“Despite the inaccuracies of the MM-GBSA methods, the method has several strengths that more accurate alchemical methods do not. It is an end-state method that allows us to calculate large changes like protein-protein complex formation, and it allows us to decompose the binding energies into van der Waals, Coulombic, polar solvation, and apolar solvation components, which aids us in understanding the molecular driving forces for complex formation. Here, we also neglect entropy, which also contributes to the error in our binding prediction. We quantify errors within our calculation by running them in quadruplicate and calculating the standard error of the mean among the four replicas. We also look at the variance within each simulation by calculating the standard deviation. These values are reported in **Supplementary Table 1**. The MM-GBSA binding energy may be calculated by running three separate simulations, or one simulation of just the complex. We then break up the complex into pieces. By running one simulation, we reduce error in our calculation, and we can see the MM-GBSA energy as a function of time.”

Third, the authors acknowledge some caveats in their simulation design (restraints, missing residues, simulation lengths, etc) but their justification remains unconvincing.

We thank the reviewer for the opportunity to clarify the simulation caveats presented in the supplemental information. These caveats do not alter the findings realized from the simulations. The caveats are presented to highlight and bring up weaknesses in our calculation. We argue that this elaboration of our simulation design adheres to the principle of promoting scientific reproducibility, and highlights factors that could impact the conclusions from our computational modeling. To support the structural work, our simulations have been used to estimate binding energies of complex formation, and to quantify the strength of residue-residue interactions among the proteins. Through the residue-residue energetic analysis, we can speak to interactions (including estimated strengths) that we would not otherwise have confidence in based on our 3.5-3.9 Å structures alone. Furthermore, our computationally determined root mean square fluctuation (RMSF) visualizations for each solved structure can be directly related to the experimental structure factors, the interaction energy values that we calculate through the residue-residue potential energy decomposition, and the observations that we draw from the dynamics of the system provide a framework for further mutagenesis experiments and binding assays. In this sense, our molecular simulation and quantitative analysis provide a means to further augment and interpret the structural data that has been obtained through cryo-EM.

We edited and added text to the convergence caveat: “3. Convergence. The simulations all seem to be sampling different regions of conformational space (evidenced by looking at the first two principal components of the combined trajectories). It is difficult to judge how long is long enough. To address convergence, we generated time series plots of the system’s potential, kinetic and total energy, as well as the MM-GBSA energies. In addition to visually looking at these plots, we calculated Block Average Standard Error of the Mean, Autocorrelation functions and the slope of the trend line. We also plotted the RMSD time series. The data seem reasonably well-behaved; these give us a sense of the internal error, and correlated trends. We see that the energies go up and down and seem to fluctuate about a mean value. The analysis also helped convince us that our equilibration procedure is reasonable. We ran four 250ns trajectories, calculated MM-GBSA binding values, and presented a statistical analysis in Supplemental Table 1, quantifying the variance within each trajectory and among the trajectories. Reassuringly, despite sampling different portions of conformational space, the binding energies are reasonably close in value, giving us confidence in the mean value (within the standard error).”

We added the following to the end of the SI: “... To support the structural work, our simulations have been used to estimate binding energies of complex formation, and to quantify the strength of residue-residue interactions among the proteins. Through the residue-residue energetic analysis, we can speak to interactions (including estimated strengths) that we would not otherwise have confidence in based on our 3.5-3.9 Å structures alone. Furthermore, our computationally determined root mean square fluctuation (RMSF) visualizations for each solved structure can be directly related to the experimental structure factors, the interaction energy values that we calculate through the residue-residue potential energy decomposition, and the observations that we draw from the dynamics of the system provide a framework for further mutagenesis experiments and binding assays. In this sense, our molecular simulation and quantitative analysis provide a means to further augment and interpret the structural data that has been obtained through cryo-EM.”

We have added the following sentence at the end of the “Molecular Dynamics simulations and analysis” section of the Methods: “See Supplementary section 3 for a discussion of caveats and Supplementary section 4 for some concluding remarks about our simulation contributions to the study.”

Despite these somewhat minor weaknesses (some of which can be addressed in a revision), the work overall is expansive and rigorous, and it represents an important progress toward a more complete understanding of signaling through the RAS/MAPK signaling pathway.

We appreciate the reviewer’s overall positive endorsement of the rigor of our work and its contribution towards understanding the MAPK signaling pathway.

REVIEWERS' COMMENTS:

Reviewer #1 (Remarks to the Author):

The revised manuscript answered the points raised by this reviewer.

Reviewer #2 (Remarks to the Author):

Dear Authors,

Thank you for taking the time to critically review your manuscript. I feel the additional material has answered my questions and questions that could not be addressed can only remain as open questions for now due to lack of appropriate data. I am therefore happy that you addressed my concerns to the best of your ability.

Reviewer #3 (Remarks to the Author):

The authors have addressed my concerns adequately. I have no further suggestions.